# Feedforward motor information enhances somatosensory responses and sharpens angular tuning of rat S1 barrel cortex neurons

Mohamed Khateb[1], Jackie Schiller[1], Yitzhak Schiller[1,2]*

[1]The Rappaport Faculty of Medicine, Technion - Israel Institute of Technology, Haifa, Israel; [2]Department of Neurology, Rambam Medical Center, Haifa, Israel

**Abstract** The primary vibrissae motor cortex (vM1) is responsible for generating whisking movements. In parallel, vM1 also sends information directly to the sensory barrel cortex (vS1). In this study, we investigated the effects of vM1 activation on processing of vibrissae sensory information in vS1 of the rat. To dissociate the vibrissae sensory-motor loop, we optogenetically activated vM1 and independently passively stimulated principal vibrissae. Optogenetic activation of vM1 supra-linearly amplified the response of vS1 neurons to passive vibrissa stimulation in all cortical layers measured. Maximal amplification occurred when onset of vM1 optogenetic activation preceded vibrissa stimulation by 20 ms. In addition to amplification, vM1 activation also sharpened angular tuning of vS1 neurons in all cortical layers measured. Our findings indicated that in addition to output motor signals, vM1 also sends preparatory signals to vS1 that serve to amplify and sharpen the response of neurons in the barrel cortex to incoming sensory input signals.

*For correspondence: y_schiller@yahoo.com

## Introduction

Rodents are equipped with an array of vibrissae on their snout (mystacial vibrissae) that serve as a highly developed tactile somatosensory organ (*Woolsey and Van der Loos, 1970*; *Carvell and Simons, 1990*; *Petersen, 2007*; *Diamond et al., 2008*; *Diamond and Arabzadeh, 2013*; *Feldmeyer et al., 2013*). To probe their environment, rodents typically produce whisking movements of their vibrissae, and palpate objects located within the vibrissae reach. The physical contact between external objects and vibrissae generate forces and vibrations at the base of the vibrissae, which are sensed by specialized mechanoreceptors located at the vibrissae pad. In turn, the sensory information collected by these mechanoreceptors is conveyed to the cortex via the VPM and PoM nuclei of the thalamus (*Petersen, 2007*; *Diamond et al., 2008*). The main cortical region receiving sensory information from the vibrissae is the somatosensory S1 barrel cortex (vS1), which is arranged somatotopically, with each vibrissa represented by a separate and discrete barrel-like region (*Izraeli and Porter, 1995*; *Petersen, 2007*; *Diamond et al., 2008*; *Bosman et al., 2011*; *Diamond and Arabzadeh, 2013*; *Feldmeyer et al., 2013*).

The vibrissae-barrel somatosensory system typically uses active sensing (*Kleinfeld et al., 2006*; *Schroeder et al., 2010*). Whisking movements are evoked by descending commands from the primary vibrissae motor cortex (vM1) (*Grinevich et al., 2005*; *Gerdjikov et al., 2013*; *Hill et al., 2011*; *Petersen, 2014*). It is still unclear whether neurons in vM1 directly drive the facial nucleus, or alternatively activate a central pattern generator (CPG) in the brainstem that in turn rhythmically drives the facial nucleus (*Grinevich et al., 2005*; *Kleinfeld et al., 2014*; *Moore et al., 2014*; *Petersen, 2014*). In addition to the descending output motor information to the brain stem, vM1 also directly sends

information to other structures in the sensory-motor vibrissae-barrel loop, and especially the vS1 barrel cortex (*Aronoff et al., 2010*; *Veinante and Deschênes, 2003*; *Mao et al., 2011*; *Feldmeyer et al., 2013*). This direct vM1 to vS1 pathway originates mostly from layer 2–3 and layer 5A pyramidal neurons in vM1 and terminates mostly on layer 5 neurons and to a lesser degree layer 2–3 neurons of vS1 (*Mao et al., 2011*). Previous functional studies have shown that vM1 inputs to vS1 carry information regarding both motor parameters of whisking, as well as more complex sensory information (*Petreanu et al., 2012*). Moreover, inputs from vM1 have been shown to modify the network state of the vS1 barrel cortex, and increase the coding reliability of complex sensory stimuli in vS1 (*Zagha et al., 2013*).

In this study, we aimed to investigate the effect of vM1 inputs on sensory processing in the barrel vS1 cortex. A major obstacle we encountered in tackling this question was the fact the vibrissae-barrel system functions as a motor-sensory loop (*Kleinfeld et al., 1999*; *Diamond et al., 2008*), and thus, vM1 can influence vS1 neurons both directly and indirectly. For example, vM1 activation can affect vS1 neuron both via direct inputs connecting the two areas and indirectly via whisking-evoked stimulation of mechanoreceptors in the vibrissae pad. To tackle this problem, we disconnected the motor-sensory loop of the vibrissa-barrel system, and dissociated activation of the motor and sensory components. We passively activated the principal vibrissa by piezo-mediated vibrissa movements or artificial whisking against sandpaper (*Szwed et al., 2003*; *Garion et al., 2014*), and independently activated vM1 with optogenetic stimulation. In all our experiments, we cut the buccolabialis branch of the facial nerve to eliminate cortical-driven whisking movements. Using this experimental paradigm, we aimed to investigate the effect vM1 activation on the response of vS1 neurons to vibrissa stimulation; characterize temporal interactions between vM1 and vS1; and examine the effect of vM1 on angular tuning of vS1 neurons.

## Results

### Pairing optogenetic vM1 activation with passive vibrissa activation

The goal of this study was to investigate the effect of vM1 on sensory processing in the vS1 barrel cortex. Under physiological conditions, the barrel-vibrissae system functions as a loop, and thus vM1 can affect the vS1 barrel cortex both directly and indirectly. To address this obstacle, we passively stimulated the principal vibrissa, and independently activated vM1 neurons using optogenetic stimulation. In the experiments we passively deflected, the principal vibrissa (Typically B2) with piezo-mediated 200 ms ramp and hold vibrissa deflection (*Bruno et al., 2003*; *Lavzin et al., 2012*), while concomitantly recording single-unit activity from neurons in the vS1 barrel cortex. We recorded the single-unit activity from neurons in different layers of the vS1 using single-shaft 16 channel silicon probe electrodes (inter contact distance of 50 μm; contacts located at depth of 300–1100 μm from the pia) (*Figure 1A*). The principal barrel was identified prior to electrode insertion using intrinsic imaging (for details see *Garion et al., 2014*), and verified by manual deflection of vibrissae during extracellular multi-unit recordings.

Typically, the 200 ms passive ramp and hold vibrissa deflection-evoked short on and off responses (*Figure 1B*). For our analysis, we measured the additional number of spikes (spike count) evoked by the stimulus compared to the pre-stimulus control value during the on (initial 100 ms of the ramp and hold stimulation) and off responses (initial 100 ms after the end of the ramp and hold stimulation). In addition, we independently activated neurons in vM1 cortex with optogenetic stimulation. We expressed ChR2 in vM1 pyramidal neurons by local injection of AAV viral vectors containing the ChR2 gene expressed under the control of the CaMKII promotor (*Figure 1C*), and ChR2 expressing neurons in vM1 were activated with 20 ms laser pulses (*Figure 1D*; for details see Materials and methods).

Extracellular recordings from vM1 during optogenetic stimulation showed that short 473 nm laser pulses effectively activated vM1 neurons (*Figure 1D*). Overall 84% of all recorded neurons in vM1 significantly increased their firing rate during 20 ms 473 nm laser pulses (109 vM1 neurons from five rats). On average vM1 neurons increased their average firing by 563% ± 84% during the pulse time. Moreover, the average latency between laser onset and increased firing in vM1 was 4.81 ± 0.43 ms (mean ± SEM, 109 neurons from five rats), with 41% of neurons showing a unimodal response.

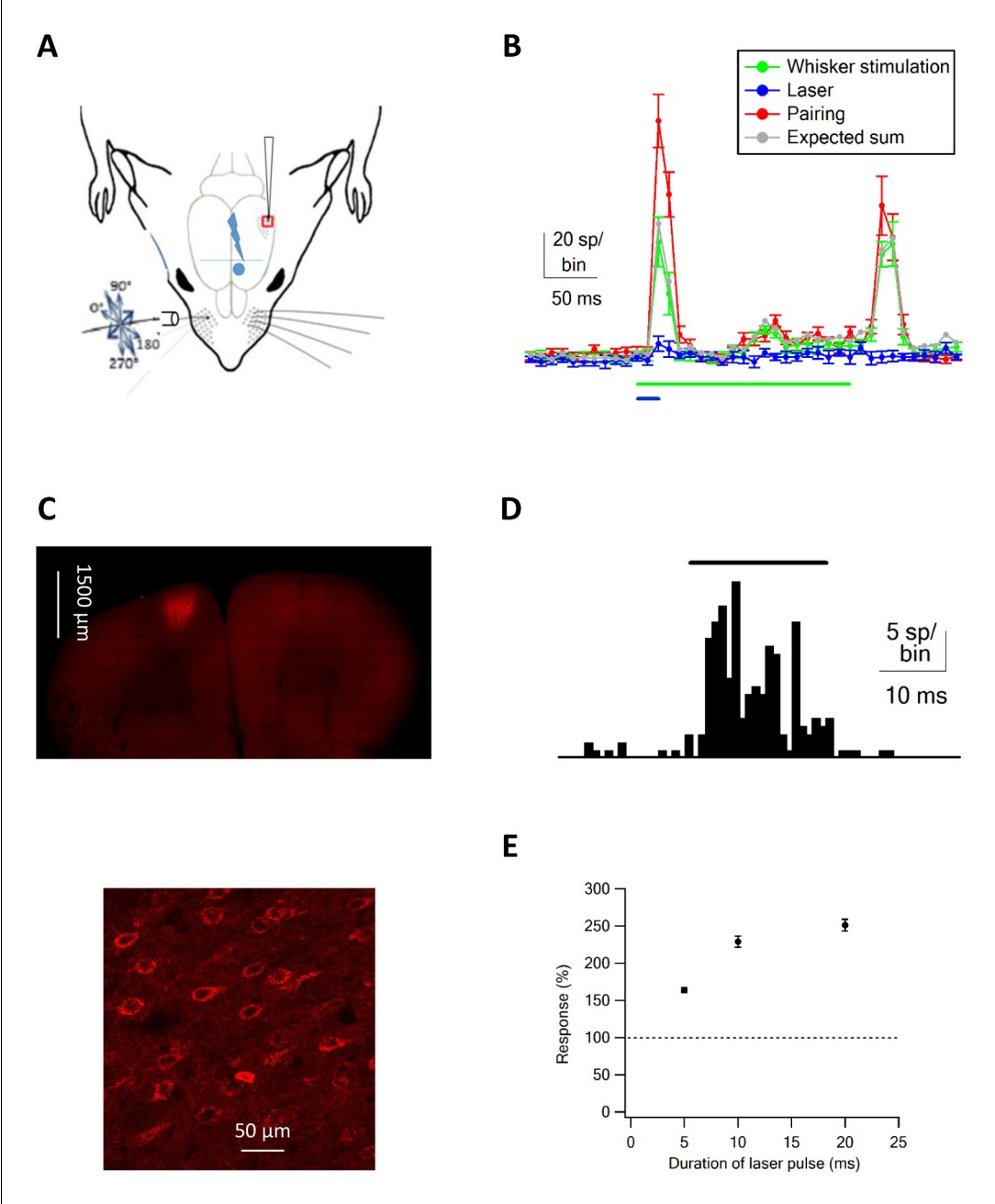

**Figure 1.** Optogenetic vM1 activation and passive ramp and hold vibrissa deflection. (A) Scheme of the experimental design with the recording electrode in vS1, optogenetic stimulation of vM1 and ramp and hold passive vibrissa deflection. (B) Peri-stimulus histograms (PSTH, mean ± SEM) recorded from a vS1 neuron during isolated vM1 optogenetic stimulation (blue), isolated passive vibrissa deflection (green), and paired vM1-vibrissa stimulation (red). In addition, the expected linear sum of the solitary vM1 stimulation and vibrissa deflection is shown (gray). (C) Fluorescence images at different magnifications of neurons co-expressing ChR2 and mCherry. (D) PSTH recorded from a vM1 neuron during optogenetic stimulation (473 nm laser pulse). (E) Average (mean ± SEM) responses recorded in vS1 neurons during optogenetic stimulation of vM1 with laser pulse duration (147 neurons in four rats). In each neurons, the spike count responses during optohgenetic stimulation were presented as percent of the spike count during the pre-stimulus control value. Later, the responses were averaged over the different neurons.

The following figure supplements are available for figure 1:

**Figure supplement 1.** Raw data of unit recordings.

**Figure supplement 2.** Raw clustering data.

*Figure 1 continued on next page*

*Figure 1 continued*

**Figure supplement 3.** Piezo bimorph movement.

When we recorded from vS1 neurons during optogenetic activation of vM1 neurons, we found small responses in vS1 barrel neurons (*Figure 1B,E*, and *Figure 2*). The magnitude of the responses recorded in vS1 neurons depended on the duration of the laser pulse, and gradually increased as the duration of the laser pulse increased from 5 to 20 ms (*Figure 1E*). Based on this data, we used 20 ms laser pulses for all our experiments.

To investigate the effect of vM1 activation on sensory processing in vS1 neurons, we applied three different stimulation conditions in random order. First, isolated vM1 optogenetic stimulation; second, isolated ramp and hold vibrissa deflection; third, paired optogenetic vM1 activation with passive vibrissa deflection. When we paired optogenetic activation of vM1 with passive ramp and hold deflection (paired vM1-vibrissa stimulation), vM1 activation supra-linearly amplified the responses in vS1 neurons. During the on response, paired vM1-vibrissa stimulation were significantly larger (by an average of 39%) than the expected linear sum of the two stimuli applied separately (*Figure 2A*, 237 neurons from seven rats). We also confirmed these findings with multi-unit analysis (*Figure 2B*). In the case of multi-unit analysis the on response to paired ramp and hold vibrissa deflection and optogenetic vM1 stimulation was 34% larger than the expected sum of the two stimuli applied separately. Moreover, we found that in 57% of individual neurons (135 out of 237 neurons) paired vM1-vibrissa stimulation was significantly larger than the expected sum of solitary vibrissa deflection and vM1 optogenetic activation.

Interestingly, a small yet significant supra-linearity was also observed during the off response despite the fact we applied the 20 ms optogenetic stimulation at the onset of the 200 ms ramp and hold vibrissa deflection. Paired vM1-vibrissa deflection stimulation yielded an average off response that was 15% larger than the expected sum of the two stimuli applied separately (expected average linear sum of 207.6% ± 11.7% compared to 238% ± 12.4% of the averaged measured paired vM1-vibrissa stimulation, $p<0.05$, 237 units from seven rats).

We also investigated the effect of pairing vM1 activation with passive ramp and hold vibrissa deflection in the different cortical layers of the vS1 barrel cortex. Our recordings extended between 300–1100 μm from the cortical surface. We divided the recorded neurons according to their putative cortical layers, based on their distance from the pia. We defined putative layers 2–3 as extending up to 600 μm from the pia, layer 4 as extending 650–800 μm from the pia, and layer 5 as extending beyond 800 μm from the pia. It is important to stress that cortical layers were determined solely by the distance of the recording contact from the cortical surface, as we could not verify cortical layers by histological criteria. Paired opto-stimulation of vM1 supra-linearly amplified the on response vS1 neurons to ramp and hold vibrissa deflection in all cortical layers recorded. Supra-linearity was largest in putative layers 2–3 and 5, and smallest in putative layer 4 neurons. No significant differences were observed between layers 2–3 and 5 (*Figure 2D*, analysis for the different layers is shown for the on response).

We further investigated the effect of vM1 activation on the response to vibrissa activation using a second passive stimulation paradigm, artificial whisking against sandpaper. With this passive stimulation paradigm, we cut the buccolabial branch of the facial nerve and stimulated the severed branch to generate artificial whisking movements at 5.5 Hz while contacting sandpaper (*Szwed et al., 2003*; *Bagdasarian et al., 2013*; *Garion et al., 2014*) (*Figure 3A*). vM1 neurons were optogenetically activated in a similar manner to the ramp and hold vibrissa deflection experiments. For our analysis, we measured the additional number of spikes (spike count) evoked by the protraction phase of artificial whisking (90 ms) as compared to the pre-stimulus baseline. We chose to concentrate on the earlier protraction phase as we opto-stimulated vM1 at the onset of artificial whisking.

Artificial whisking against sandpaper-evoked spikes during the protraction and retraction phases (*Figure 3B*). Similar to the ramp and hold stimulation paradigm optogenetic co-activation of vM1 supra-linearly amplified the response of vS1 neurons to artificial whisking against sandpaper. On average, the response evoked by paired vM1 optogenetic activation with artificial whisking against

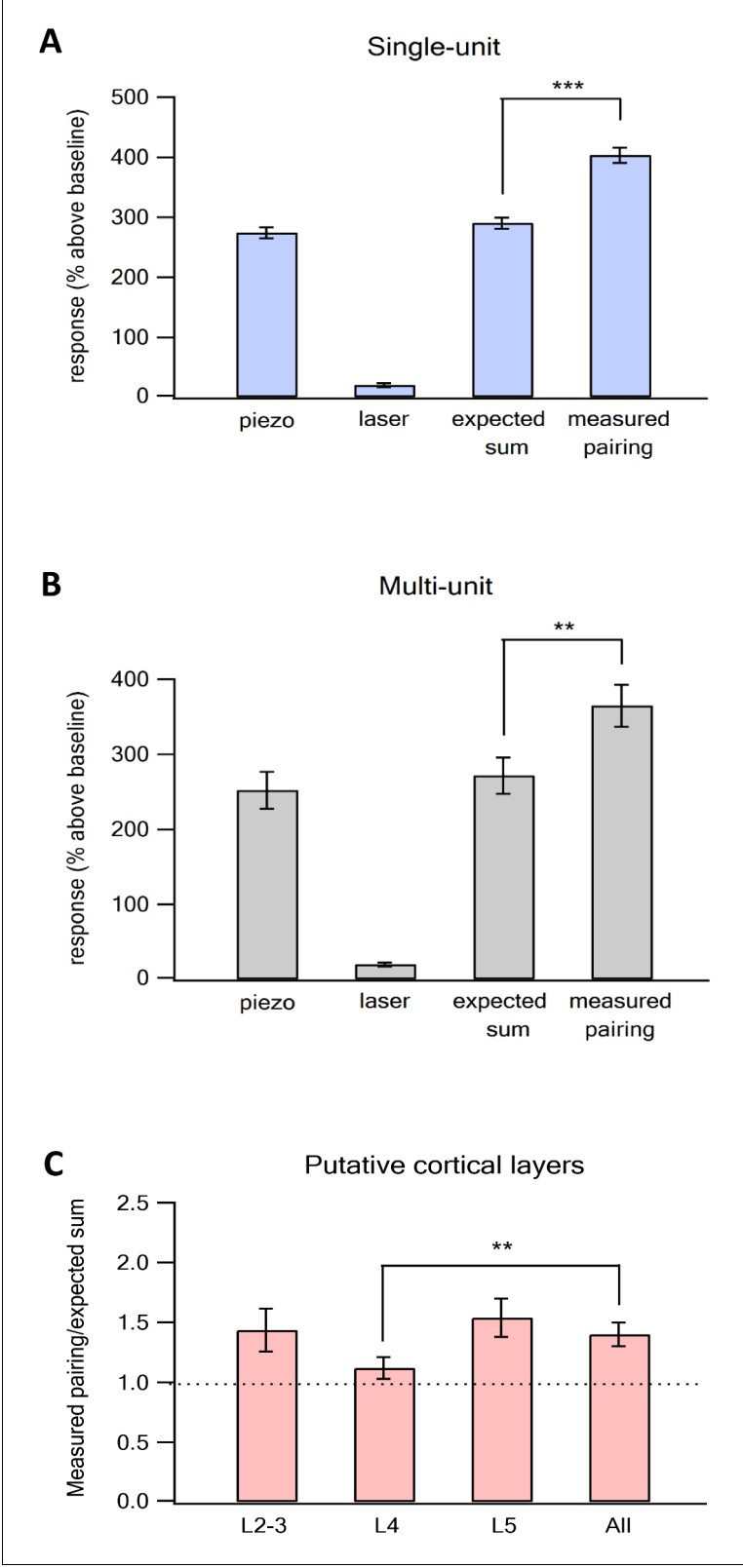

**Figure 2.** Supra-linear summation of paired optogenetic vM1 activation and passive ramp and hold vibrissa deflection-averaged results. Average (mean ± SEM) single-unit response (presented as percent above the pre-stimulus control activity) of vS1 neurons to the On response during the different stimulation conditions, isolated vM1 optogenetic stimulation, isolated passive vibrissa deflection, expected linear sum of the solitary vM1

*Figure 2 continued on next page*

*Figure 2 continued*

stimulation and vibrissa deflection and paired vM1-vibrissa stimulation presented for single-unit analysis (**A**) and multi-unit analysis (**B**), (237 units from 91 electrode contacts in seven rats). Note that the recorded activity during paired vM1-vibrissa stimulation was significantly larger than the linear sum of the response to vM1 activation and passive ramp and hold vibrissa deflection applied separately for both single- and multi-unit analyses. (**C**) Average (mean ± SEM) supra-linearity of the paired vM1-vibrissa stimulation response (measured/expected linear sum) presented for all neurons and for neurons in the different putative neocortical layers (2–3, 4 and 5). Thirty-five neurons from putative layers 2–3, 79 neurons from putative layer 4 and 123 neurons in putative layer 5. ***p<0.001 and **p<0.01 using the student's t-test.

sandpaper was 42% larger than the expected linear sum of the two individual responses applied separately (*Figure 3B,C*). When we calculated the ratio of the measured versus expected responses for individual neurons we found that 69% of neurons (164 out of 241 neurons) had measured paired vM1-vibrissa stimulation that were significantly larger than the expected linear sum of isolated vibrissa deflection and vM1 optogenetic activation. Similar to ramp and hold vibrissa stimulation, pairing vM1 activation with artificial whisking supra-linearly amplified the response in all recorded neocortical layers (*Figure 3D*).

## Temporal roles governing the interactions between vM1 and vibrissa sensory inputs

In contrast to physiological conditions, our experimental paradigm allowed us to activate the vibrissae and vM1 in a temporally independent manner. We utilized this capability to investigate the temporal rules governing the effect vM1 activation on the response of vS1 barrel neurons to passive vibrissa stimulation. In these experiments, the vibrissa and vM1 were stimulated independently with different time delays ranging between −100 ms and +50 ms (time difference calculated by subtracting the onset time of vibrissa stimulation from the onset time of the laser pulse, namely the onset of vibrissa stimulation was defined as time 0 ms). For these experiments, we passively stimulated the principal vibrissa by either ramp and hold vibrissa deflection (*Figure 3A*) or artificial whisking against sandpaper (*Figure 3B*). When vM1 was optogenetically activated 0–50 ms prior to passive vibrissa stimulation the paired response was significantly larger than that evoked by isolated passive vibrissa stimulation (*Figure 3*). In contrast, when vM1 was activated 75–100 ms before or 20–50 ms after passive vibrissa stimulation the combined response did not significantly differ from the isolated vibrissa stimulation. The largest facilitating effect of vM1 activation was observed when the onset of the optogenetic vM1 activation preceded the onset of vibrissa stimulation by 20 ms (*Figure 3*). These findings were observed for both ramp and hold vibrissa deflection and artificial whisking against sandpaper (*Figure 3A,B*).

It is important to stress that in our experiments the time difference between vM1 optogenetic activation and vibrissa stimulation was in fact the time difference between the onset of vibrissa stimulation and onset of optogenetic stimulation, without taking into account transmission and synaptic delays, nor the exact timing of slip and stick events for artificial whisking.

## The effect of vM1 activation on angular tuning of neurons in the vS1 barrel cortex

Previous studies have reported that neurons in the vS1 barrel cortex show a preference for the direction in which the vibrissa is deflected (angular tuning) (*Bruno et al., 2003*; *Andermann and Moore, 2006*; *Kremer et al., 2011*; *Lavzin et al., 2012*). In the previous section, we have shown that paired vM1 activation supra-linearly enhances the response of vS1 barrel cortex neurons to vibrissa stimulation. We next set out to examine whether in addition, paired vM1 activation also effects angular tuning of neurons in the vS1 barrel cortex. In these experiments, we recorded the response of vS1 barrel neurons to passive ramp and hold deflection of the principal vibrissa to eight different directions with and without paired optogenetic activation of vM1. vM1 optogenetic activation was applied at the optimal temporal time window (laser onset preceding vibrissa deflection by 20 ms). In these experiments, we randomly alternated the direction of vibrissa deflection and whether the laser was activated or not. Consistent with previous results, we found that angular tuning in neurons

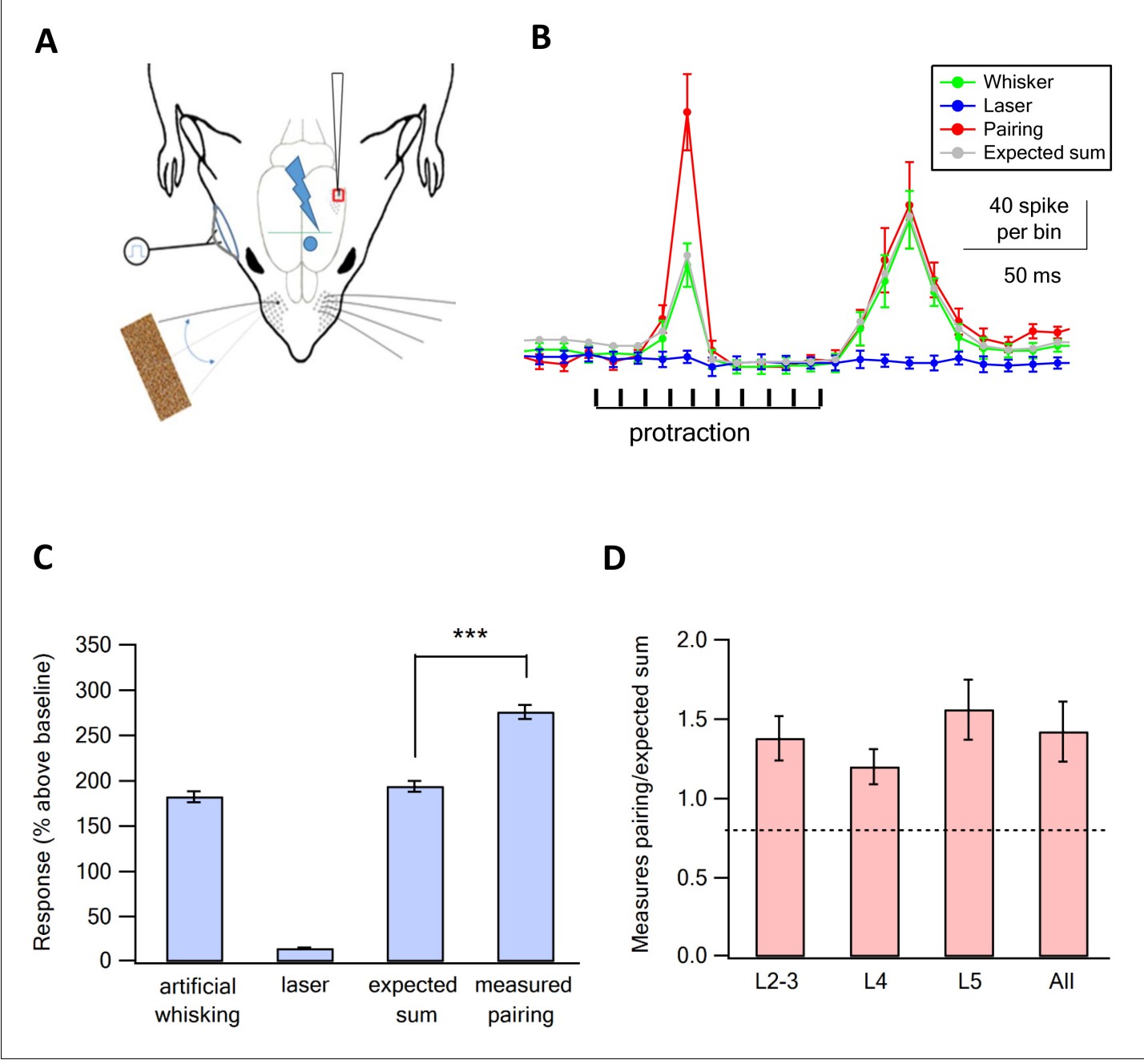

**Figure 3.** Supra-linear summation of paired optogenetic vM1 activation and artificial whisking against sandpaper. (A) Scheme of the experimental design with the recording electrode in vS1, optogenetic stimulation of vM1 and artificial whisking against sandpaper by electrically stimulating the buccolabialis nerve. The timing of individual nerve stimulation and protraction phase is marked in black. (B) Peri-stimulus histograms (PSTH, mean ± SEM) recorded from a vS1 neuron during solitary vM1 optogenetic stimulation (blue), solitary artificial whisking against sandpaper (green), and paired vM1-vibrissa stimulation (red). In addition, the expected linear sum of the solitary vM1 stimulation and artificial whisking is shown (gray). (C) Average (mean ± SEM) responses (presented as percent above the pre-stimulus control activity) of vS1 neurons to the different stimulation conditions (294 neurons from six rats). Note that the recorded activity during paired vM1-vibrissa stimulation was significantly larger than the linear sum of the response to solitary vM1 activation and artificial whisking. ***p<0.001 with the paired student's t-test. (D) Average (mean ± SEM) supra-linearity of the paired vM1-vibrissa stimulation response (measured/expected linear sum) presented for all neurons and for neurons in the different putative neocortical layers, 76 neurons from putative layers 2–3, 80 neurons from putative layer 4 and 138 neurons in putative layer 5.

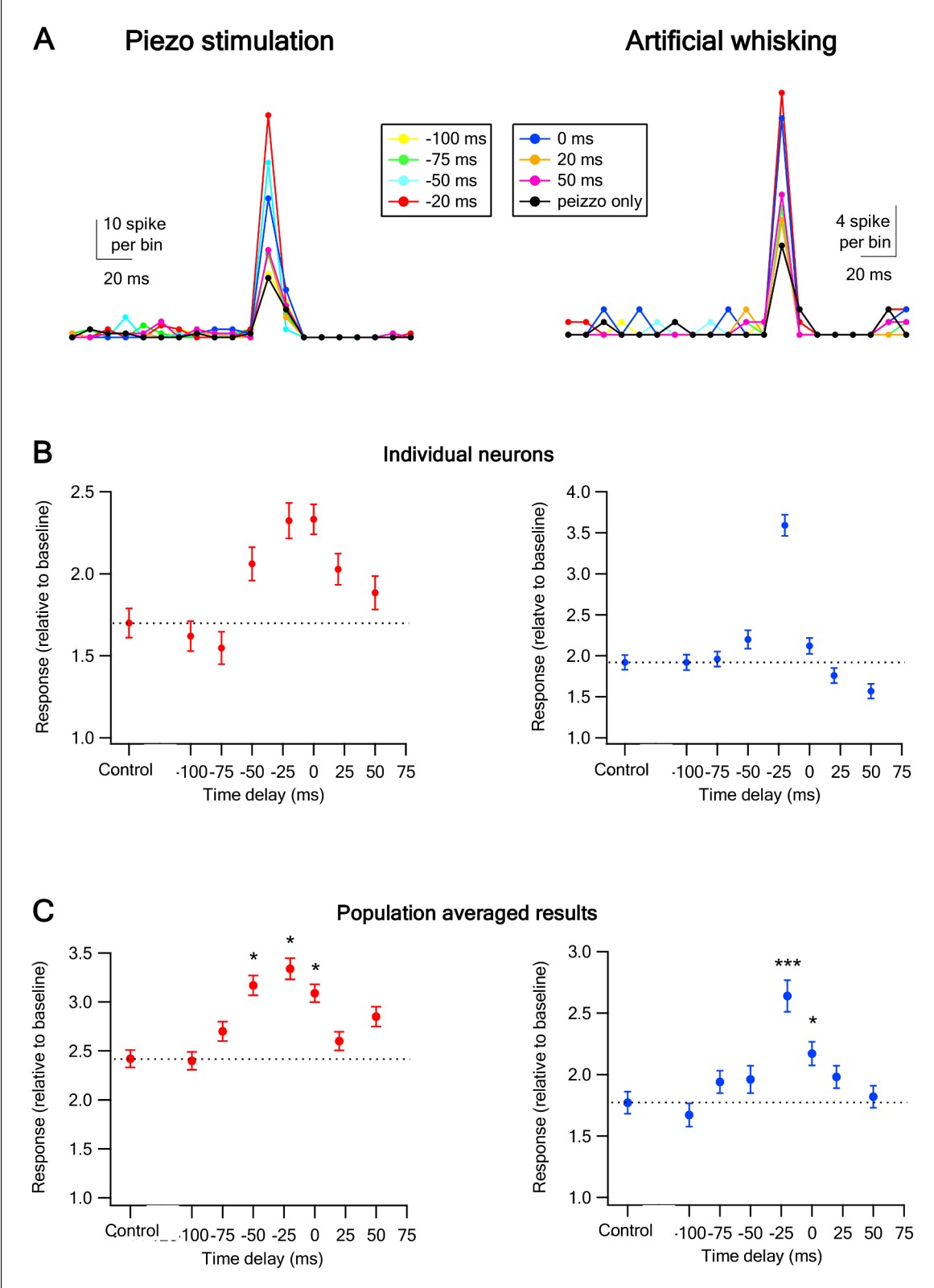

**Figure 4.** Temporal rules governing the interactions between of vM1 activation and vibrissa stimulation on the response of vS1 barrel cortex neurons. (**A**) The peak of the peri-stimulus histograms (PSTH) recorded from individual vS1 neurons during paired vM1-vibrissa stimulation with different time lags between the vM1 optogenetic activation (20 ms laser pulse) and vibrissa stimulation (−100 ms up to +50 ms). The time lags were calculated by subtracting the onset time of vibrissa stimulation from the onset time of vM1 optogenetic activation. The data are presented for two different individual

*Figure 4 continued on next page*

*Figure 4 continued*

neurons during passive ramp and hold vibrissa deflection (left panel) and artificial whisking against sandpaper (right panel). (**B**) The response (mean ± SEM) of two different individual vS1 neurons to paired vM1-vibrissa stimulation applied with different inter-stimuli time lags between the optogenetic and vibrissa deflection. In the left panel, the vibrissa was stimulated with a ramp and hold vibrissa deflection, and in the right panel, the vibrissa was stimulated with artificial whisking against a P320 sand paper. (**C**) The average (mean ± SEM) response of all recorded vS1 neurons to isolated passive vibrissa stimulation (control) and paired vM1-vibrissa stimulation applied with different inter-stimuli time lags. For both **B** and **C**, the data are presented for passive ramp and hold vibrissa deflection (left panels) and artificial whisking against sandpaper (right panels). 238 neurons from six rats for vibrissa deflection experiments; 334 neurons from seven rats in the artificial whisking experiments; *p<0.05, ***p<0.001.

is located in layers 2–3 and layer 4 (*Bruno et al., 2003*; *Andermann and Moore, 2006*; *Kremer et al., 2011*; *Lavzin et al., 2012*). We further found that layer 5 neurons in the vS1 barrel cortex also show similar angular tuning. We quantified the percent of neurons with significant angular tuning, and found that 62% ± 8% of all recorded neurons showed significant angular tuning, with no significant differences between putative layers (significant angular tuning was defined by significant difference at the 0.05 level for comparison of the responses to the preferred angle and to the three least preferred angles).

When we paired optogenetic stimulation of vM1 with vibrissa deflection to different angles, we found that the angular tuning of neurons was sharpened. Sharpening of angular tuning by optogenetic activation of vM1 is demonstrated in six different individual vS1 neurons (*Figure 5*, *Figure 5—figure supplement 1*). To quantify the effect of vM1 activation on angular tuning of barrel vS1 neurons, we calculated the selectivity index (SI) of angular tuning of the preferred direction (maximal response in the preferred angle/average response to all angles) for each neurons with and without vM1 optogenetic activation. In these experiments, we found that paired vM1 optogenetic activation significantly increased the average SI for angular tuning in vS1 barrel cortex neurons (*Figure 6A*), which resulted from a right shift of the SI value histogram by vM1 activation (*Figure 6B*). Interestingly, we observed a significant increase in the SI of the preferred direction in both 50–60 day old and 90–100 day old rats (*Figure 6—figure supplement 1A*). Furthermore, as expected, while the SI of the preferred direction increased, the SI values of the worst three angular directions decreased by vM1 activation, although this reduction did not reach statistical significance (*Figure 6—figure supplement 1B*). It is important to stress that although the SI to the worst directions decreased, in many cases the activation of vM1 increased the absolute response to the worst angles (see examples in *Figure 5*, and *Figure 5—figure supplement 1*).

To overcome potential technical problems associated with single-unit analysis, we repeated our analysis for angular tuning using multi-unit activity. Similar to single- unit activity multi-unit activity, again demonstrated that vM1 optogenetic activation significantly sharpened angular tuning of vS1 neurons (*Figure 6A*). The amplificatory effect of vM1 activation. The fact we observed sharpening of angular tuning with multi-unit analysis is consistent with the previously reported spatial mapping of angular tuning in the barrel cortex (*Kremer et al., 2011*).

To further quantify the effect of vM1 optogenetic activation on angular tuning, we calculated the vector sum of the responses to vibrissae stimulation in the eight different angles (see Materials and methods for details) with and without vM1 optogenetic activation. We found that similar to the effect on SI, vM1 optogenetic activation significantly increased the amplitude of the vector sum (*Figure 6C*). On average, the amplitude of the vector sum increased by 29% ± 7%.

A significant sharpening of angular tuning by paired vM1 activation was observed in all recorded neocortical layers (layers 2–5), with no significant differences between the recorded putative layers (*Figure 6D*). The majority of neurons retained their preferred angular direction during opto-stimulation of vM1. The probability for retaining the preferred angular direction during paired vM1 activation was larger in neurons with higher SI during solitary unpaired vibrissa stimulation, with 70.9% of neurons with SI greater than two retaining their preferred angular direction after paired vM1 activation (*Figure 6E*). Moreover, neurons that retained their preferred angular direction showed a significantly greater degree of sharpening of their angular tuning curve by vM1 optogenetic activation, as compared to neurons that changed their preferred angular tuning during paired vM1 activation (*Figure 6F*).

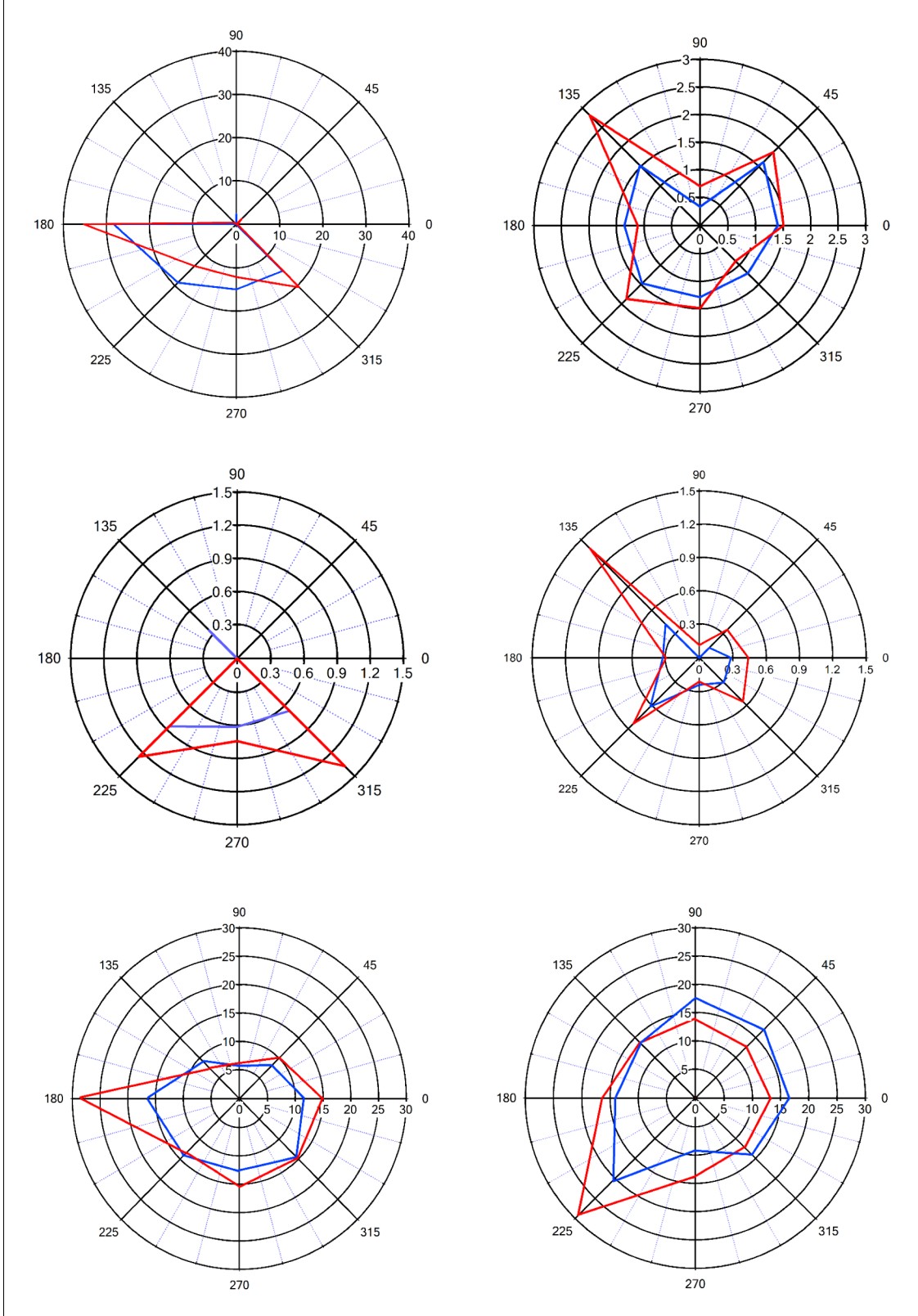

**Figure 5.** Sharpening of the angular tuning of vS1 by paired vM1 activation: examples of individual neurons. Polar plots of six individual vS1 neurons during isolated ramp and hold vibrissa deflection (blue) and paired vM1-vibrissa stimulation (red). In these experiments, the principal vibrissa was randomly deflected to eight different directions (0°,43°,90°, 135°, 180°, 225°,270°, 315°) with and without paired vM1 optogenetic activation. Vibrissae were deflected in the different directions by two pairs of galvanometers. Upper four panels are from 50–60 day old rats, and the lower two panels are

*Figure 5 continued on next page*

*Figure 5 continued*

from 90–100 day old rats. Upper left panel-putative layer 2–3, upper right panel putative layer 5, middle left panel-putative layer 4, middle right panel putative layer 5, lower left panel-putative layer 5, lower right panel putative layer 2–3. Note, sharpening of the angular tuning curve following paired vM1 optogenetic activation.

The following figure supplement is available for figure 5:

**Figure supplement 1.** The effect of vM1 activation on the angular tuning of individual neurons.

## Discussion

In this study, we set out to investigate the direct effects of vM1 activation on sensory processing in neurons of the vS1 barrel somatosensory cortex. The main findings of this study include: (1) vM1 optogenetic activation supra-linearly amplified the response of vS1 barrel neurons to passive vibrissa stimulation. This supra-linear amplification effect of vM1 activation was observed in all neocortical layers recorded (layers 2–5 300–1100 µm from the pia), and recurred in two different passive stimulation paradigms, ramp and hold vibrissa deflection and artificial whisking against sandpaper. (2) The maximal effect of vN1 on the response of vS1 neurons to vibrissa sensory inputs occurred when the onset of vM1 activation preceded vibrissa activation by 20 ms. Smaller yet significant effects were also observed when the onset of vM1 activation preceded vibrissa stimulation by 50 ms or when vibrissa stimulation and vM1 activation occurred simultaneously. This temporal relationship recurred for the two passive stimulation paradigms we examined. The physiological time delay between vM1 and vS1 activation is not fully known. Yet previous studies have shown that the activity of both vM1 and vS1 is phased locked with exploratory rhythmic whisking movements in rats (*Ahrens and Kleinfeld, 2004*), and vM1 neurons are probably critical in rhythmically driving whisking (*Carvell et al., 1996*; *Kleinfeld et al., 2002*; *Ahrens and Kleinfeld, 2004*; *Brecht, 2004*; *Brecht et al., 2004*) (3). We found that in addition to supra-linearly amplification of the response of vS1 barrel neurons to vibrissa stimulation, activation of vM1 also significantly sharpened the angular tuning in these neurons.

### vM1 connections to the vS1 barrel cortex

Previous studies have shown that vM1 neurons directly innervate neurons in the vS1 barrel cortex (*Izraeli and Porter, 1995*; *Veinante and Deschênes, 2003*; *Diamond et al., 2008*; *Aronoff et al., 2010*; *Mao et al., 2011*; *Feldmeyer et al., 2013*). The strongest monosynaptic connections between vM1 and vS1 exist between neurons in layers 2–3 and layer 5A of vM1 and layer 5A and 5B neurons in the vS1 barrel cortex (*Mao et al., 2011*). Interestingly, reciprocal connections also exist between layers 2–3 and 5A neurons in vS1 and vM1 neurons (*Hooks et al., 2013*). A subset of vM1 axons reach layer 1 of the vS1 barrel cortex, and carry both motor information regarding different vibrissa movement parameters and more complex sensory information regarding contact of vibrissae with objects, and spatial localization of objects (*Petreanu et al., 2012*). Moreover, vM1 inputs innervating tuft dendrites of vS1 layer 5 pyramidal neurons critically participate in dendritic spike initiation when vibrissae actively touched objects in a location and angle-dependent manner (*Xu et al., 2012*).

In this study, we show that short (20 ms) optogenetic stimulation of vM1 neurons significantly increased firing of vS1 barrel neurons. These findings are in agreement with the results of a recent study performed in awake behaving rodents that reported activation of vS1 barrel neurons by optogenetic stimulation of vM1 neurons (*Zagha et al., 2013*). Moreover, *Zagha et al. (2013)* also reported that optogenetic activation of vM1 resulted in context-dependent changes in the network state of the barrel cortex and increased reliability of responses to complex sensory stimuli.

Although previous studies have established the existence of direct connections between vM1 and vS1, and have demonstrated functional implications of this pathway (*Petreanu et al., 2012*; *Xu et al., 2012*; *Zagha et al., 2013*), our study adds novel yet unknown information regarding the effects of vM1 inputs on vS1 barrel neurons. First, we show that motor information from vM1 neurons and sensory information from the vibrissae summate supra-linearly in vS1 barrel neurons. Second, we defined the temporal rules governing the interactions between incoming vM1 and vibrissa sensory input information in vS1 neurons. We found maximal vM1-vibrissa interactions occurred

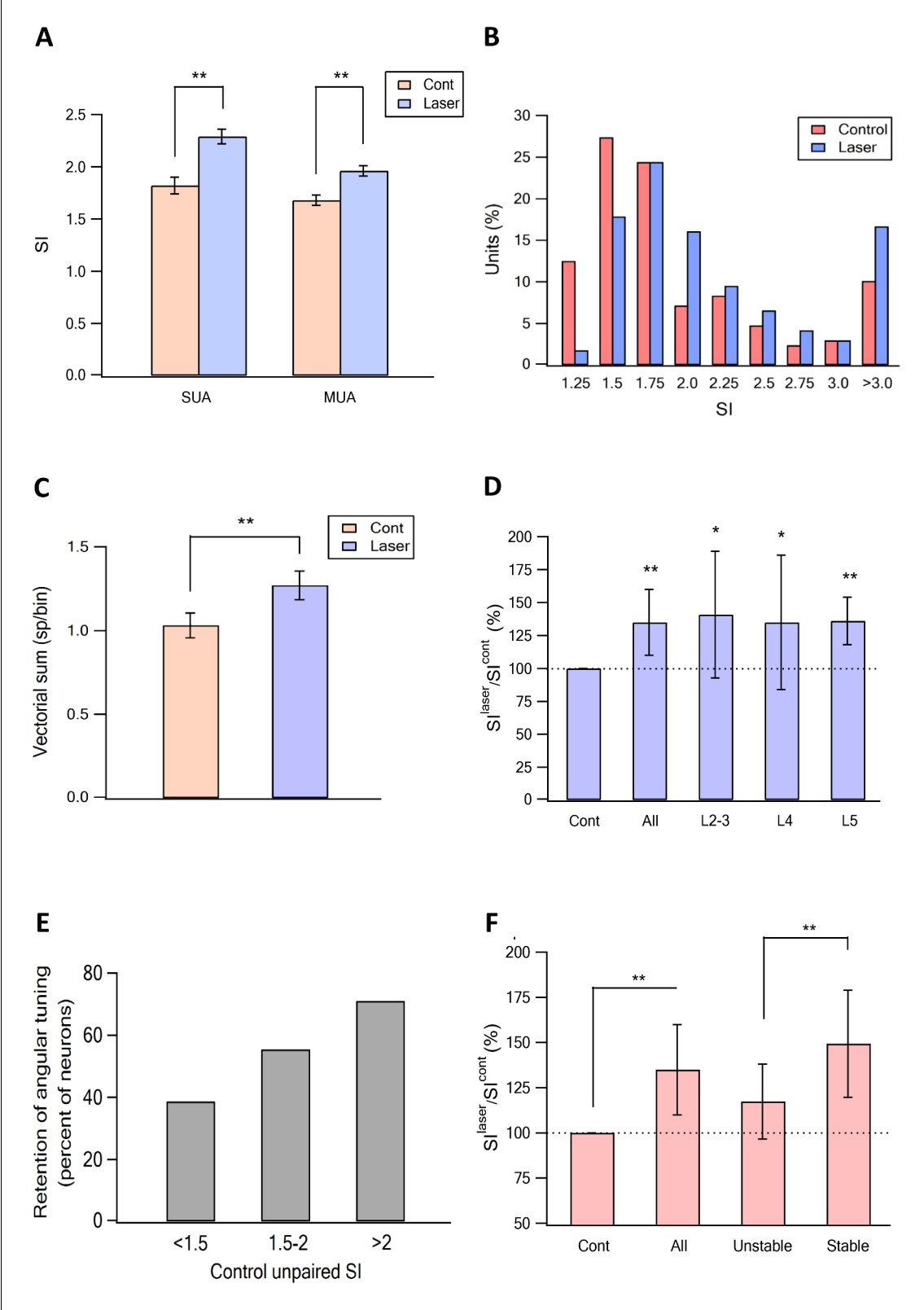

**Figure 6.** Sharpening of the angular tuning of vS1 by paired vM1 activation: averaged results. (**A**) The average (mean ± SEM) ratio between the SI calculated for the preferred angular direction for isolated vibrissa deflection (red) and paired vibrissa deflection with vM1 optogenetic activation (blue). The results are presented for single- (SUA) and multi-unit (MUA) analysis. The results are shown for all neurons examine (304 neurons from 11 rats). Note the significant increase in the SI following pairing with vM1 activation for moth SUA and MUA. (**B**) The SI magnitude histogram of unit for isolated for
*Figure 6 continued on next page*

*Figure 6 continued*

isolated vibrissa deflection (red) and paired vibrissa deflection with vM1 optogenetic activation (blue). Note that paired vM1 optogenetic stimulation resulted a right shift of the histogram. (C) The effect of vM1 optogenetic activation on the average amplitude (mean ± SEM) of the vector sum. Vector sum analysis was performed on the same data set presented in panels A and B (204 neurons from 11 rats). **p<0.01. (D) The average (mean ± SEM) ratio between the SI calculated for isolated vibrissa deflection and paired vibrissa deflection with vM1 optogenetic activation. The results are shown for all neurons examine, and for the different putative cortical layers (25.7% putative layer 2–3 neurons, 20% putative layer 4 neurons and 54.3% putative layer five neurons). *p<0.05, **p<0.01. In cases the SI for isolated vibrissa deflections were compared with paired vibrissa deflections and vM1 optogenetic activation using the paired student's t-test. (E) Percent of neurons that retained the same SI with and without vM1 optogenetic activation as a function of the control SI value (SI < 1.5, SI = 1.5–2 and SI > 2). (F) The average (mean ± SEM) ratio between the SI value recorded without ($SI^{cont}$) and with ($SI^{laser}$) vM1 optogenetic activation in all recorded neurons and in neurons which retained the preferred angle with and without optogenetic activation (stable), and in neurons that changed their preferred angle after vM1 activation (unstable). **p<0.01.

The following figure supplement is available for figure 6:

**Figure supplement 1.** The effect of vM1 activation on angular tuning.

when vM1 activation preceded vibrissa stimulation by 20 ms. This optimal temporal window for vM1-vibrissa input interactions suggested that vM1 sent primarily feedforward information regarding the planned vibrissa movements, rather than feedback information originating in the sensory motor loop. Finally, we show that vM1 activation, not only amplified the response of vS1 barrel neurons to incoming vibrissa sensory inputs, but also sharpened their angular tuning curve. Taken together, it seems vM1 inputs participated in increasing both the sensitivity and specificity of barrel cortex neurons to incoming sensory information from the vibrissae. The direct effect of vM1 on vS1 neurons may underlie the difference between passive and active sensing in the barrel vibrissa somatosensory system.

## Possible mechanisms underlying vM1-mediated response amplification and sharpening of angular tuning in vS1 barrel neurons

In our study, we did not directly address the cellular mechanisms underlying the effects of vM1 inputs on the response of vS1 neurons to incoming vibrissa stimulation. Several different potential cellular and network mechanisms can explain our findings. First, is a non-linear effect on the axonal initiation zone. Specifically with this mechanism, excitatory vM1 inputs depolarize vS1 neurons, and as a result, concomitant vibrissae-evoked EPSPs will generate a larger number of action potentials in the axonal initiation zone. Second, are non-linear dendritic amplification mechanisms. Previous studies have shown that dendrites of neocortical excitatory neurons can support generation of local dendritic sodium, calcium and NMDA spikes (*Stuart et al., 1997*; *Larkum et al., 1999*, *2009*; *Schiller et al., 2000*; *Polsky et al., 2004*; *London and Häusser, 2005*; *Branco et al., 2010*; *Lavzin et al., 2012*; *Xu et al., 2012*; *Harnett et al., 2013*; *Major et al., 2013*; *Smith et al., 2013*; *Palmer et al., 2014*; *Cichon and Gan, 2015*). It is possible that pairing vM1 inputs with inputs carrying vibrissa sensory information results in initiation of local dendritic spikes vS1 neurons. Consistent with this possibility are the results of several previous in vivo studies that reported initiation of dendritic spikes in tuft dendrites of vS1 layer-5 pyramidal neurons in response to vM1 and M2 activation (*Xu et al., 2012*; *Manita et al., 2015*). Third, are local network mechanisms. For example, vM1 inputs may selectively innervate VIP inhibitory inter neurons, as described for the medial prefrontal inputs to the auditory cortex (*Lee et al., 2013*; *Pi et al., 2013*). In this case, vM1 inputs will activate VIP inter neurons, which will inhibit other inhibitory inter neurons, and in turn secondary excite pyramidal neurons in vS1.

## Possible functional significance of vM1 mediated response amplification and sharpening of angular tuning in vS1 barrel neurons

The barrel vibrissa system uses active sensing to palpate objects in the near vicinity of the rodents head. Typically, vM1, the motor cortex of the barrel vibrissa system, is responsible for generating the whisking movements of the mystacial vibrissae. vM1 can drive whisking either by directly activating lower motor neurons in the facial nucleus, especially during protraction movements, or alternatively, by activating a brain stem CPG located in close proximity to or within the Botzinger complex.

In the latter case the CPG drives the facial nucleus to generate rhythmic movements (*Grinevich et al., 2005*; *Haiss and Schwarz, 2005*; *Gerdjikov et al., 2013*; *Moore et al., 2013*; *Petersen, 2014*; *Sreenivasan et al., 2015*). In addition to the brainstem output motor commands, vM1 also directly innervates the vS1 barrel cortex. These inputs convey both motor parameters of whisking, as well as more complex sensory information regarding contacting the object and object localization (*Petreanu et al., 2012*; *Harnett et al., 2013*).

Functionally, there are several scenarios for how the vM1 to vS1 pathway participates in sensory computations. First, vM1 inputs can convey a simple 'attentional cue' to prepare vS1 barrel cortex neurons for the incoming sensory information from the vibrissae. Regarding this possibility, it is interesting to note that previous studies have shown that increased attention and especially activation of the frontal eye field area increases the sensitivity and specificity of cortical neurons in sensory visual regions to different visual stimuli (*McAdams and Maunsell, 1999*; *Reynolds and Chelazzi, 2004*; *Noudoost et al., 2010*). The second possibility is that vM1 inputs convey a motor efference copy that can be used for computing the location as well as shape and texture of objects by comparing the expected (vM1) with the measured (thalamo-cortical) vibrissa movements (*Diamond et al., 2008*; *Curtis and Kleinfeld, 2009*; *Kleinfeld and Deschênes, 2011*). Third, multiple inputs converge onto the barrel motor cortex, thus vM1 inputs can serve to convey top-down dynamical modifications of sensory processing in the primary vS1 cortex (*Xu et al., 2012*; *Zagha et al., 2013*; *Manita et al., 2015*), see also *Squire et al. (2013)* regarding the visual system). Regardless of the exact computational role vM1 inputs play in sensory processing in vS1 neurons, the significant effects vM1 inputs have on vS1 neurons may underlie the importance of active sensing in the somatosensory system. Further studies in awake behaving rodents are needed to distinguish between these three possibilities and decipher the functional role of M1 inputs in sensory processing in somatosensory processing.

## Materials and methods

### Surgical preparation for recording in anesthetized rats

We conducted experiments in accordance with NIH and institutional standards for the care and use of animals in research, and received the approval of our institutional animal ethics committee (protocol 007-01-2014). P50-60 Wistar rats were anesthetized by intra-peritoneal injection of Urethane (20% dissolved in normal saline). Prior to surgery lidocaine (2%) was applied locally over the scalp, and the buccolabialis branch of the facial nerve, which innervates muscles in the vibrissae pad, was exposed and severed at its proximal segment. The skull was exposed and well cleaned in order to identify important anatomical landmarks such as the midline, bregma and lambda. A craniotomy (2–3 mm$^2$) was drilled over the vS1 barrel cortex (2.5 mm posterior to bregma, 4.5 mm lateral to the midline), and a well surrounding the craniotomy was constructed using dental cement. In addition, a short metal pole was glued to the skull rostral to the dental cement well and later used to hold the rats head in place. After the dental cement was constructed, the dura matter was carefully removed over a small area (less than 1 mm$^2$) to allow access to the cortex. We minimized the area of exposed cortical surface in order to reduce brain pulsation and damage during the hours of the recordings. The well surrounding the craniotomy was filled with aCSF to cover the exposed neocortical surface to prevent the brain from drying. Body temperature was carefully maintained at 36–37°C using a heating pad (FHC, Montana, USA).

### Optical intrinsic imaging

To identify the principal barrel, we initially performed optical intrinsic imaging as previously described (*Lavzin et al., 2012*; *Garion et al., 2014*). The cortical surface was illuminated and imaged through the thinned skull. We used 650 nm LED to illuminate the cortex, and light absorbance images were acquired with a Q-cam CCD camera (QImaging, British Columbia, Canada). Individual vibrissae were deflected using a pair of galvanometers at 10 Hz for a 2 s duration, to identify the location of the barrel representing the stimulated vibrissa. Sensory stimulation and data acquisition were controlled via an isolated pulse stimulator (model 2100, A-M systems, Washington, USA) using custom home made software written in Matlab (MathWorks, Massachusetts, USA). The location of the principal barrel was later used to guide electrode insertion.

## Electrophysiologcal unit recordings

Electrophysiological recordings were performed simultaneously from multiple neurons in layers 2–5 of the vS1 barrel cortex using silicone multi-contact probes (NeuroNexus, Michigan, USA). Our silicone probes (A 1 × 16 MEA) were composed of a single-shaft electrode with 16 recording contacts arranged along a vertical line and separated by 50 micrometers. Electrodes were inserted to pre-mapped principal barrel using a stereotactic micro-manipulator (TSE, Bad Homburg, Germany). Electrodes were slowly lowered until the electrode tip reached approximately a depth of 1000–1100 μm from the pial surface. After the electrode was inserted into the barrel cortex, we verified the identity of the principal vibrissa by recording multi-unit activity during manual deflection of the vibrissa. During the experiments, we trimmed all vibrissae aside from the principal vibrissa.

Electrophysiological data was acquired with the 16-channel ME-16 system and MC Rack software (Multi channel systems, Reutlingen, Germany). The recorded data were initially amplified (X1000), filtered at 0–25 kHz and stored in the computer, and the analysis was performed primarily offline. In addition, to allow for online monitoring of the unit activity during the experiments the data was filtered online at 1–5 kHz and displayed. For the offline analysis, the raw recorded data were replayed and filtered at 1–5 kHz to obtain the unit activity (*Figure 1—figure supplement 1*). Later single-units were sorted using the OFS offline spike sorter (Plexon, Texas, USA), and the sorted spike trains were further analyzed using the Neuroexplorer software (Nex Technologies, Alabama, USA) and home written software in MatLab (MathWorks, Massachusetts, USA). The results were presented as peri-stimulus histograms (PSTH), and the increase in spike count during stimulation, defined as the additional number of spikes evoked during stimulation stimuli above the number of spikes during the control pre-stimulus baseline (Spike count during stimulation-Spike count during a similar baseline control).

For offline sorting, we initially detected events with an amplitude >3.5 SD of the baseline value. These events served for multi-unit analysis (MUA). For single-unit analysis, we further sorted the thresholded events using semi-automatic clustering algorithms, followed by manual verification and correction of these clusters, if needed. Clusters were accepted as single-units if all the following criteria were met: (1) the waveform shape remained consistent and stable throughout recording (verified by the 'Sort-Quality Vs Time' analysis in the OFS software). Moreover, units were excluded in case the average amplitude or half width of unit changed significantly (ANOVA test) between the initial and last 20% of recorded spikes. (2) Firing rate was >0.5 Hz to allow for adequate sampling. (3) Inter-spike interval (ISI) was >2 ms to reflect the absolute refractory period of neurons. (4) ISI distribution showed a smooth exponential-like curve. (5) Finally and most importantly, statistical criterion of p<0.05 (multivariate ANOVA) of cluster separation. *Figure 1—figure supplement 2* shows sorting parameters of 10 individual units. The average signal to noise (SNR) value of our units was 11.2 ± 0.4 (range of 7–15, *Rousche et al., 1999*)

In addition, we performed cross-correlation analysis between units recorded for all adjacent contact pairs, and excluded the unit from one of the electrodes (with the smaller amplitude) in case the peak of the cross-correlation was >0.9 (2 ms time bin) to avoid recording of the same unit with two contacts. In addition, we excluded all units that showed >0.9 value of the cross-correlogram in more than one electrode pair to exclude noise.

All the averaged results are presented as the mean±SEM values, and statistical testing was performed using the paired and unpaired student's t-test. The source code for the Matlab homemade software are enclosed in supplementary files (*Source code 1*).

To verify the recording location, at the end of the experiment a fluorescent dextran (florescent dextran-A solution of 2 mM fluorescent dextran Alexa-488 or Texas Red; Invitrogen, USA) was injected into the electrode tract using a pressure injector. Later the rat was sacrificed, and trans-cardially perfused with 4% Para-Formaldehyde for histological processing.

## Vibrissa stimuli

We used three different passive stimulation paradigms applied to the principal vibrissa:

1. Artificial whisking paradigm as previously described (*Derdikman et al., 2006*; *Garion et al., 2014*). In this stimulation paradigm, we exposed and stimulated the buccolabialis branch of the facial nerve to induce artificial whisking movements. The buccolabialis nerve was cut, and its distal end mounted on a pair of bipolar tungsten electrodes. We applied a train of 10

protraction-retraction cycle applied at 5.5 Hz. For each protraction. Ten bipolar rectangular electrical pulses (0.5–4.0 V, 40 μs duration) were applied through an isolated pulse stimulator (A360, WPI) at 100 Hz to produce vibrissa protraction, followed by a passive vibrissa retraction. The stimulation magnitude was adjusted to the minimal value that reliably generated the maximal possible movement amplitude. All other vibrissae except the principal vibrissa were cut. The tip of the principal vibrissa contacted a piece of P320 sandpaper (2 Cm$^2$), which the vibrissa brushed against during the artificial whisking. Artificial whisking was visually monitored under a stereo microscope to ensure proper movements of the vibrissa. We repeated the stimulation at each condition 132 times, and divided the repetitions to three equal blocks (each containing 44 repetitions). Blocks of the different stimulation conditions are applied in random order during the experiments.

We repeated the stimulation in each condition 132 times, divided to three equal blocks (each containing 44 repetitions). Blocks of the different stimulation conditions were applied in random order during the experiments.

2. Passive ramp-and-hold stimulation of the principal vibrissa. With this stimulation paradigm, the principal vibrissa was rapidly deflected (1300°/second) with a single ceramic piezoelectric bimorph for a period of 200 ms (*Simons, 1983*; *Wilent and Contreras, 2004*; *Lavzin et al., 2012*; *Garion et al., 2014*). To avoid ringing of the vibrissa during the rapid deflection phases, we generated a sigmoidal onset and offset of the ramp and hold pulses. To generate the sigmoidal onset and offset of the pulse we first, generated a sloped onset and offset phase lasting five millisecond. Second we applied a forward low-pass filter on the stimulus waveform using the following equation: $X_n=X_{(n-1)}+(X_n-X_{(n-1)})*SF$, where SF designates the smoothening factor.

3. 3. Third, we used a consecutive backward low-pass filter on the stimulus wave form using the following equation: $X_n=X_{(n+1)}+(X_n-X_{(n+1)})*SF$.

We tested SF ranging from 0.03–1, and for our experiments we used a SF = 0.03, resulting in an effective rise/fall duration of approximately 10–20 ms (*Figure 1—figure supplement 3*). Piezo bimorph deflections were controlled via a National Instruments board (PCI 6713), using custom routines written in MatLab. To monitor and calibrate vibrissa movements, and confirm lack of distortion or ringing of the stimulated vibrissa the deflection was monitored using a laser displacement sensor (LD1605-2; Micro-Epsilon OptoNCDT 1700) (*Lottem and Azouz, 2009*; *Lavzin et al., 2012*; *Garion et al., 2014*), and a high-speed camera (1000 fps) (Flare, 4M180MCL, 4 Megapixel, Dalsa Xcelera-x4-CL, IO industries) the Streams six acquisition software (IO industries) and homemade analysis software written in Matlab (MathWorks, NA) (*Garion et al., 2014*). *Figure 1—figure supplement 3A* presents single traces of piezo bimorph movements during ramp and hold pulses generated with different smoothening factors (0.03–1). *Figure 1—figure supplement 3B* shows the average (mean±SEM) peak-to-peak ringing amplitude during ramp and hold stimulation pulses. Note that under our stimulation conditions (SF = 0.03) overshoot and ringing was minimal.

4. Passive ramp-and-hold deflection of the principal vibrissa to eight different directions to identify the angular tuning of neurons. The principal vibrissa was deflected in the different directions with 200 ms ramp-and-hold stimuli using two perpendicular pairs of ceramic piezoelectric bimorphs (For calibration of the piezoelectric bimorphs see previous section, *Lavzin et al., 2012*; *Garion et al., 2014*). Altogether, the principal vibrissa was deflected to eight different directions separated by 45° (0°, 45°, 90°, 135°, 180°, 225°, 270°, 315°), and delivered at 0.5 Hz to prevent steady-state adaptation of vibrissa-evoked responses. We repeated the ramp and hold vibrissa deflection to each angle direction 50 times with the vM1 laser off and 50 times with the vM1 laser on. The repetitions for each direction with and without vM1 laser activation were divided into two equal blocks and the different deflection blocks were applied in a random order.

Similar to the case of ramp and hold vibrissa deflection to a single direction we calibrated the perpendicular bimorphs, and made sure that no ringing and distortions occurred during stimulation we used a sigmoid function for the onset and offset piezo movements, resulting in an effective rise/fall duration of approximately 10–20 ms, and monitored movements of the vibrissa using both a laser displacement sensor (LD1605-2; Micro-Epsilon OptoNCDT 1700) and visual vibrissa tracking with a high-speed camera (Flare, 4M180MCL, 4 Megapixel, Dalsa Xcelera-x4-CL, IO industries at 1000 fps) (See above).

To quantify angular tuning we used two different methods:

1. *Selectivity index (SI).* The SI reflected the ratio between the response to the preferred angle and the average response to all angles. For each neuron, we calculated the Selectivity index (SI) using the following formula:

$$SI = Preferred / \left( \sum Ri^*0.125 \right)$$

With Rpreffered, designating the response at the preferred angle, and $\sum Ri$ designating the sum of responses obtained in each one of the eight directions. In some case, we also calculated the SI of non preferred angles, by replacing Rmax with response to the relevant angle.

2. *Vector sum.* For each neuron, we plotted the response to each angle as a vector on a 2-demminsional plane. In turn, the eight vectors obtained for the different stimulation angles were summed to a single vector. Thus, the overall response was represented as a single vector with an amplitude and direction (*Mazurek et al., 2014*).

## Viral vector injection and optogenetic stimulation

To transfect pyramidal neurons of vM1 with channelrhodopsin 2 (ChR2), a craniotomy was drilled over the vM1 region (2 mm anterior to bregma and 1.25 mm lateral to the midline) under Isoflurane anesthesia and local anesthetic injection in P25-30 rats. A solution (500 nl) containing the ChR2 expressing viral vector (AAV2.1.CAMKIIa-hChR2(H134)-mCherry.WPRE.hGH, UNC viral core facility, North Carolina, USA) was injected into the vM1 region via a small craniotomy. ChR2 expressed in excitatory neurons by using the promotor CaMKII. Viral vby a micromanipulator (Sutter Instruments, California, USA) and connected to glass pipettes with tip diameters of 30–50 um. To allow for ChR2 expression in the target pyramidal neurons electrophysiological recordings were performed 4–6 weeks after viral vector injections. At the end of each experiment, the location and extent of ChR2 expression were verified histologically. Typically, viral vectors were injected at two depths of 300 and 500 μm. These injections typically resulted in ChR2 expression in a cortical region with vertical and horizontal extents of 500–600 (*Figure 1*).

1. To activate vM1 during the experiments (performed 4–6 weeks after viral vector injections to allow for ChR2 expression), a craniotomy was re-opened over the vM1 and vS1 region, and a dental cement well was constructed around the craniotomy. The dental cement well was filled with artificial cerebrospinal fluid (aCSF) to prevent drying of the cortical surface during the recording. To activate ChR2 channel, short 473 nm light pulses were generated with a laser (OEM laser pulses, Utah, USA) controlled by an isolated pulse generator (STG4, multichannel system, Germany). The 473 nm laser was connected to a light guide that was held 10–20 mm above the neocortical surface. Control experiments revealed that in the absence of ChR2 multiple repeated laser pulses do not affect the activity of neurons in vM1. We repeated the stimulation in each condition (including paired optogenetic stimulation and vibrissa stimulation) 132 times, divided to three equal blocks (44 repetitions), and the blocks were applied in random order.

In control rats (intact un-severed buccolibial nerve), optogenetic stimulation of vM1 resulted in whisking movements (n = 5), functionally confirming the location of vM1. Upon cutting the buccolabial nerve, optogenetic stimulation of vM1 showed whisking movements in all rats tested, including the five rats in which vM1 optogenetic activation yielded whisking prior to cutting the buccolabial nerve.

## Additional information

### Funding

| Funder | Author |
| --- | --- |
| Israel Science Foundation | Jackie Schiller |
| Adelis Foundation | Yitzhak Schiller |
| Prince Neurodegeneration Center | Yitzhak Schiller |

The funders had no role in study design, data collection and interpretation, or the decision to submit the work for publication.

## Author contributions

MK, Data curation, Formal analysis, Investigation; JS, Resources, Supervision, Funding acquisition; YS, Conceptualization, Formal analysis, Supervision, Funding acquisition, Validation, Methodology, Writing—original draft, Writing—review and editing

## Author ORCIDs

Yitzhak Schiller, http://orcid.org/0000-0002-6106-4580

## Ethics

Animal experimentation: This study was performed in strict accordance with the recommendations in the Guide for the Care and Use of Laboratory Animals of the National Institutes of Health. All of the animals were handled according to approved institutional animal care and use committee (IACUC) protocols of the technion.(protocol 007-01-2014) All surgery and experiments were performed under sodium urethane anesthesia, and every effort was made to minimize suffering.

# Additional files

## Supplementary files

• Source code 1. Matlab homemade software.

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
