## [Decision Letter]

[Editors’ note: a previous version of this study was rejected after peer review, but the authors submitted for reconsideration. The first decision letter after peer review is shown below.]

Thank you for submitting your work entitled "Feedforward motor information enhance somatosensory responses and sharpen angular tuning of S1 barrel cortex neurons" for consideration by *eLife*. Your article has been favorably evaluated by Gary Westbrook (Senior Editor) and three reviewers, one of whom, Sacha Nelson, is a member of our Board of Reviewing Editors. Our decision has been reached after consultation between the reviewers. Based on these discussions and the individual reviews below, we regret to inform you that your work will not be considered further for publication in *eLife*.

Each of the reviewers felt the underlying issues were important and the major findings were potentially interesting, but each also had significant (often overlapping) concerns with the rigor of the methods, the main points being

1) the rigor of spike sorting and quality of the recordings

2) the details of stimulus timing and stimulus control, and

3) clarity surrounding the issue of response variability and stability – both generally and as it relates to the significance of the angular tuning and enhancement thereof.

It is the policy at *eLife* to reject manuscripts if the reviewers feel that additional work is needed to support the conclusions of the paper, and if in the editors' opinions this work is extensive enough to require more than about 2 months. We would be open to considering a new revised manuscript if the authors want to try to address the issues outlined above in a new manuscript. We wish to stress, however, that it might be difficult for the authors to raise the manuscript over the bar, because of the many concerns raised.

*Reviewer #1:*

The authors use multi-site extracellular recording in anesthetized rats to study the influence of M1 projections on sensory responses to whisker stimulation in S1. This is a well studied area and has been the subject of a handful of recent high profile papers. The present study makes an important contribution to this field by breaking the normally recurrent sensory-motor loop so as to be able to study the effect of motor cortex activity in the absence of the indirect effects of the movement that activity would normally produce. To do this, they sever the motor nerve to prevent centrally generated whisking and then stimulate the whiskers passively or by causing artificial whisking by directly stimulating the nerve, while at the same time stimulating the motor cortex ontogenetically. They find that there is a feed-forward facilitation of sensory responses and that this facilitation is strongest for the preferred stimulus, thereby enhancing stimulus specificity. Although there have been other demonstrations of circuits that might produce this facilitation, the sensory and temporal features of the facilitation have not been studied under conditions in which it can be cleanly isolated as here.

1) The manuscript is in need of substantial editing for usage and for clarity. I have tried to catch as many of the errors as I can, but it extends to being unsure of precisely what procedures were followed and what some of the figures depict.

2) The inclusion criteria with respect to response stability are not clear and may be inappropriate. It is stated that two parameters were used: 1) the baseline pre-stimulus frequency – but this is not justified and no criteria are given, and 2) "…conditions was repeated twice…and neurons in which the responded differentially to the two stimulation blocks were excluded." This is not clear and no definition of difference is given, but treating the responses as an n=2 repetitions and excluding based on differences is not valid without some statistical definition of repeatability. Along these lines, the variability of the responses to different directions of stimulation does not seem to have been adequately assessed. Tuning curves for individual neurons should have error bars so that the claim of enhanced tuning can be assessed statistically on a neuron-by neuron basis.

3) The results are quantified solely with respect to the on response. Off responses should also be shown (examples of the full-time course of individual responses with and without M1 stimulation should be shown) so as to reveal the kinetics of facilitation and to perhaps justify the quantification of on responses only.

4) It is not clear what is different about Figure 1 (39% larger) and 1G all (73.5% larger). The same lack of clarity applies to Figure 2. Is Figure 3 a single example neuron? If so, should a) make it clear that the response on the left and the response on the right are two individual neurons and b) the variability of the response across trials should be indicated. Figure 4 should also include measures of response variability. In Figure 5, the asterisk should be defined and exactly which groups were being compared should be stated clearly (e.g. in the Methods or in the legend or text). If the pooled controls are being compared to individual layers (i.e. the layers don't have their own separate control groups – which would not make sense but which is somewhat implied by the structure of the figure) a correction for multiple comparisons must be used.

5) The authors should show some example histology that convinces the reader that the assignment to layers is accurate.

*Reviewer #2:*

The supralinear main effect observed in this paper is undeniably interesting.

Major:

I) Stimulus control is essential for this experiment:

1) Perpendicularly-arrayed bimorphs are notorious for ringing and distortion. I would like quantification of the amplitude and velocity of motion for all 8 directions for the piezo's used, and most importantly the ringing they demonstrated both for the first deflection and the other deflections in a train. Such quantification and description is essential.

2) I'm confused by the description of stimulation in the Results (200 ms ramp and hold stimuli) and in the Methods (100 ms).

II) I am skeptical about the 50 micron MEAs from neuronexus finding any single units of demonstrable selectivity in layers II/III. I certainly am not impugning the author's basic credibility (I don't think they are lying!) but getting well-isolated units in those layers, or layer IV especially, with such probes is quite hard. Without the virtue of a tetrode configuration or the use of smaller diameter contacts the claim of single unit identification is particularly suspect.

I also found the following criteria to feel a bit arbitrary and wasn't sure what it meant:

"For our analysis, we excluded all unstable neurons, with stability determined using two parameters. First, the baseline pre-stimulus frequency. Second, for every experimental paradigm at least one of the stimulation conditions was repeated twice during the experiment, and neurons in which the responded differentially to the two stimulation blocks were excluded."

First, what does the phrase "the baseline pre-stimulus frequency" have to do with stability? Second, multi-unit recordings can easily show run-by-run stability and, in fact, are potentially more stable than units for such recordings.

So, I'd like to see the following to give me confidence:

1) Application of autocorrelogram exclusion criteria.

2) Quantification of spike feature stability across the entire run (whether or not the response properties changed), and a discussion of which spike features were used.

3) Demonstration of the cluster quality for some number of units –.I don't want to make this infinite, but as I said I'm skeptical, so at least a fair number of examples of stability in layers II/III units would be appreciated (say, 10 such examples from 10 experiments or so).

Moderate:

I) In the 56 neurons recorded in vMI, what was the latency to firing following optogenetic stimulation, and was this a bimodal distribution (implying directly opto activated and synaptically opto activated populations).

II) The authors should quantify the relative effect of stimulation on the optimal direction angle versus the worst direction angle, whatever these might have been. Obviously, there must be enhancement on average across randomly associated directions or else the main effect of Figure 1 couldn't stand: The sharpening obviously suggests, though, a significant different between best and worst, etc.

*Reviewer #3:*

Khateb et al. report on modulation of whisker-stimulation evoked spike rates in rat barrel cortex by optogenetic activation of projections from motor cortex. To avoid confounding effects by sensory reafference activation they prevented initiation of whisker movements by cutting the motor branch of the facial nerve. For two stimulation paradigms under anesthesia they find that pairing of motor cortex activation with whisker stimulation leads to supralinear spiking responses in barrel cortex neurons across all layers, most prominently in L2/3 and L5. This effect depended on the relative timing of the two inputs, peaking when motor cortex activation shortly preceded the whisker stimulation. In addition, they report a sharpening of tuning to the angular direction of stimulation upon pairing.

How a cortical sensory area such as the barrel cortex integrates the various input streams it receives, is a fundamental, most relevant question. Modern tools, especially optogenetic precise control of specific pathways as applied here, now enable addressing this question in vivo. This study therefore is timely and presents highly original data. The finding that neuronal spiking activity in barrel cortex is amplified by activation of the M1-S1 pathway is highly significant, even though the underlying potential mechanisms are not investigated here, as the authors admit. The results regarding changes in angular tuning I find less convincing, given that this feature is a complex, debated issue and little data are provided here. Overall, this is a nice study, making an important contribution to signal processing in the barrel cortex. The manuscript would benefit, however, from a more detailed and extensive presentation of (raw) data and a more in-depth treatment of the crucial issue of relative input timing. Find my specific comments below.

1) Introduction, last paragraph: Why only 'partially disconnected'? Isn't the motor arm entirely uncoupled? Was this actually verified by confirming the absence of whisker movements following vM1 activation after the nerve cut?

2) Subsection “Electrophysiological unit recordings” and figures. The authors should provide some raw electrophysiology data, so that one can better judge the quality of data. What were the noise levels? How well could spike sorting be performed and on how many channels? How well could spike waveforms be separated and were there any putative fast-spiking units present? Do the reported spike rates represent changes in spike rate? It would also be interesting to see the LFP responses for the paired stimulation paradigms.

3) Furthermore: what were the respective baseline spike frequencies and why (and how) were these used to exclude data from the analysis (subsection “Electrophysiological unit recordings”, third paragraph). I also do not understand the motivation for excluding units that did not show a consistent response pattern upon repeating the stimulation paradigm. What type of responses did these cells actually show, in how far did they deviate from a nonlinear summation results? Why should one set such bias?

4) Artificial whisking paradigm (subsection “Whisker stimuli”, Figure 2, Figure 3). How exactly was the optogenetic stimulation paired with motor activation pulses? Was it a 20-ms light pulse at the beginning of the 10 pulses to the nerve? How large were the whisker protractions induced by this protocols? The spiking response appears only 20-ms long, how come? What happened in the remaining 80 ms of protraction/retraction cycle? In the eighth paragraph of the subsection “Pairing optogenetic vM1 activation with passive whisker activation” it is stated that spikes were evoked during protraction and retraction phase, but I can't see that. More details are needed here.

5) Subsection “Viral vector injection and optogenetic stimulation”, first paragraph. How precisely was the vM1 hit with the injections and the light stimulation? Was this verified by measuring whisker movements (or other movements?) by optogenetic vM1 stimulation before the facial nerve was cut?

6) I find the timing experiments very interesting but I am not sure how to interpret the peak revealed at -20 ms. This is a time scale where the conduction delays etc. play an important role. So how was the timing exactly defined (onset of piezo drive and onset of LED illumination, I presume)? Was there any dead time for mechanical stimulation of the whisker considered? In particular, how was the timing defined for the artificial whisking stimulus? Were the whisker movements (presumably stick-slip events) monitored and their timing analyzed? Were any axonal conduction times and synaptic delays taken into account? Obviously, these questions are important to understand what the real timing difference at the integrating neurons in barrel cortex might be. Supplementary whole-cell recordings could be very helpful here, to quantify when exactly inputs from both pathways actually arrive with these stimulation paradigms.

7) Angular tuning in barrel cortex apparently is a complicated matter and among other things seems to especially depend on age (Kremer study). In addition, the housing conditions (use of whiskers) may affect the outcome. The age of rats used here is just in between the ages when no angular tuning was observed and when it later was established. I find the examples in Figure 4 not convincing, as the pure whisker-evoked responses appear relatively untuned and the responses enhanced by optogenetic vM1 activation mostly display multiple peaks (often in orthogonal directions). Cells from what layer are actually shown in Figure 4? It might be helpful to show distributions of the absolute SI values for the different stimulation protocols.

[Editors’ note: what now follows is the decision letter after the authors submitted for further consideration.]

Thank you for submitting your article "Feedforward motor information enhance somatosensory responses and sharpen angular tuning of S1 barrel cortex neurons" for consideration by *eLife*. Your article has been favorably evaluated by Gary Westbrook (Senior Editor) and three reviewers, one of whom, Sacha B Nelson (Reviewer #1), is a member of our Board of Reviewing Editors, and another one is Fritjof Helmchen (Reviewer #3).

The reviewers have discussed the reviews with one another and the Reviewing Editor has drafted this decision to help you prepare a revised submission.

All three reviewers were pleased with the addition of new data and new analyses addressing the issue of spike sorting. The reviewers remain concerned about several issues outlined below. In addition, the manuscript could still benefit from additional editing.

Essential revisions:

1) One reviewer states: "I appreciate the addition of details on how the stimulator was calibrated-these are all the right approaches and tools. I can guarantee that using these parameters, that unless they use a specific compensatory algorithm, they almost certainly cannot get ringing under 5% of stroke magnitude, and I doubt it can be under 10%. I want the following included in the final manuscript → An actual analysis of the mean amplitude of the ringing. Saying you did not see it does not reflect quantification of the effect. Say how large in angle of vibrissal base motion and at what frequency the ring is on average across trials and across the different stimulators used." The reviewer notes that this is likely to have an effect on the direction selective responses.

2) A second reviewer felt that the documentation of the degree of direction selectivity was inadequate. They did not feel the statement "Consistent with previous results we found angular tuning in neurons.…" (subsection “The effect of vM1 activation on angular tuning of neurons in the vS1 barrel cortex”, first paragraph) was well supported. They felt that the criteria for when a cell's response is considered well tuned were not clear and that the SI as calculated made comparisons across neurons with very different response amplitudes difficult. It is suggested that the authors "confirm their interpretation with an alternative, more robust analysis method of direction-tuning, based on the mean response vector not the Rmax (Kremer et al. 2011; Mazurek et al., Front Neural Circ 2014). And provide a statistical argument for their statement 'we found angular tuning'."

3) One of the reviewers also notes that "laser stimulation of vM1 alone in essentially all cases did not evoke spiking activity at all (except perhaps for Figure 1?). Thus, the 'supralinear' effect essentially consists in an upregulation or facilitation of the sensory-evoked response in vS1, which could be simply explained by additional pre-depolarization mediated by the M1-to-S1 projections, given vM1 is stimulated at the right time briefly before the sensory stimulus. While the authors mention this simple explanation (vM1 projection fibers helping vS1 neurons to reach the – nonlinear – spike threshold), they only refer to it as 'additional cellular mechanism' in the third paragraph of the subsection “Possible mechanisms underlying vM1 mediated response amplification and sharpening of angular tuning in vS1 barrel neurons”. Their primary 'attractive potential mechanism' of dendritic amplification (in the first paragraph of the aforementioned subsection) in my view is, however, largely speculative." The authors should consider toning down their use of 'supralinear responses' and may consider using terms like 'facilitation' or 'modulation.' At the very least they should give more equal weight to the simplest interpretation of these effects.

---

## [Author Response]

[Editors’ note: the author responses to the first round of peer review follow.]

*Each of the reviewers felt the underlying issues were important and the major findings were potentially interesting, but each also had significant (often overlapping) concerns with the rigor of the methods, the main points being*

*1) the rigor of spike sorting and quality of the recordings*

*2) the details of stimulus timing and stimulus control, and*

*3) clarity surrounding the issue of response variability and stability – both generally and as it relates to the significance of the angular tuning and enhancement thereof.*

1) Following the concerns raised by all reviewers regarding the rigor of our spike sorting we redefined stringent spike sorting criteria, and reanalyzed all our data according to these criteria. 2) To overcome inherent difficulties associated with single unit spike sorting we performed multi-unit analysis of our data. This analysis, which is much less prone to technical inaccuracies, confirmed all the main findings of the manuscript. 3) We performed additional experiments, most important of which are a new series of angular tuning experiments in more mature rats (90-100 day old). 4) We added data regarding the variability of our recordings. 5) We added raw experimental data. We present raw filtered (1-5 KHz) recording traces and spike sorting of 10 individual electrodes with 26 individual single units. 6) We greatly expanded the description of various technical issues regarding the stimulation paradigms and timing of stimulation.

*Reviewer #1:*

*[…] 1) The manuscript is in need of substantial editing for usage and for clarity. I have tried to catch as many of the errors as I can, but it extends to being unsure of precisely what procedures were followed and what some of the figures depict.*

Following the reviewer’s comment we have extensively edited the manuscript.

*2) The inclusion criteria with respect to response stability are not clear and may be inappropriate. It is stated that two parameters were used: 1) the baseline pre-stimulus frequency – but this is not justified and no criteria are given, and 2) "…conditions was repeated twice…and neurons in which the responded differentially to the two stimulation blocks were excluded." This is not clear and no definition of difference is given, but treating the responses as an n=2 repetitions and excluding based on differences is not valid without some statistical definition of repeatability. Along these lines, the variability of the responses to different directions of stimulation does not seem to have been adequately assessed. Tuning curves for individual neurons should have error bars so that the claim of enhanced tuning can be assessed statistically on a neuron-by neuron basis.*

We thank the reviewer for his comments on this subject. Upon re-reading the manuscript, we also saw that our inclusion and exclusion criteria of units were not sufficient. For the revised manuscript, we re-defined criteria for including units in our analysis, and re-analyzed all our data according to these new and better-defined criteria.

Specifically, we initially recorded the multi-unit activity (MUA) by recording events with an amplitude >3.5 SD above baseline from the filtered raw data (1-5 KHz). From this data we next sorted single unit activity using the OFS offline spike sorting software from Plexon. Sorting was initially performed by semi-automatic algorithms, and later verified and corrected manually. We accepted clusters as single unit if they met all the following criteria: 1) the waveform shape remained consistent and stable throughout recording. This was verified by the "Sort-Quality Vs. Time" analysis in the OFS software. Moreover, we excluded unstable units in case the average amplitude or half width of units changed significantly (ANOVA test) between the first and last 20% of recorded spikes. 2) The firing rate was >0.5 Hz to allow for adequate sampling. 3) The inter-spike interval (ISI) was >2 ms to reflect the absolute refractory period of neurons. 4) The ISI distribution showed a smooth exponential-like curve. 5) Finally and most importantly statistical criterion of p<0.05 (multivariate ANOVA) of cluster separation.

These points were added to the revised manuscript (subsection “Electrophysiological unit recordings”). In addition to the revised manuscript we added examples of raw data traces (Figure 1—figure supplement 1) as well as raw clustering data (Figure 1—figure supplement 2).

In addition, in the revised manuscript we took a second approach to overcome the inherent difficulties associated with spike sorting. We repeated all our analysis for multi-unit data (threshold >3.5 SD above baseline). We found the similar to single unit analysis, multi-unit analysis also showed that vM1 optogenetic activation both supra-linearly amplified the responses of whisker stimulation, as well sharpened angular tuning. Thus, our multi-unit data verified the single unit results. The multi-unit data was added to the revised manuscript (subsection “Pairing optogenetic vM1 activation with passive whisker activation”, fifth paragraph and subsection “The effect of vM1 activation on angular tuning of neurons in the vS1 barrel cortex”, third paragraph; Figure 2 and Figure 6). Note that in the case of angular tuning we added date from 6 additional 90-100 day old mice.

Additionally, following the reviewer's comments, we added error bars to the curves of all individual neurons presented in the manuscript (revised Figure 1, Figure 3–Figure 5).

*3) The results are quantified solely with respect to the on response. Off responses should also be shown (examples of the full-time course of individual responses with and without M1 stimulation should be shown) so as to reveal the kinetics of facilitation and to perhaps justify the quantification of on responses only.*

We chose to concentrate on the "on response" as the duration of the whisker stimulus is 200 ms, and thus the "OFF response" occurs approximately 180 ms after the optogenetic pulses ended.

Following the reviewer’s comments we also analyzed the off responses to ramp and hold stimulation. To our surprise, we found that pairing of ramp and hold stimuli with optogenetic activation of vM1 also resulted in supra-linear amplification of the "off response", although the magnitude of the amplification was smaller. The additional data regarding the "off response" is added the revised manuscript (subsection “Pairing optogenetic vM1 activation with passive whisker activation”, sixth paragraph).

*4) It is not clear what is different about Figure 1 (39% larger) and 1G all (73.5% larger). The same lack of clarity applies to Figure 2.*

The difference in the values result from different calculation methods of the same data. While in Figure 1 and Figure 2 we initially measured the response for individual units and averaged the response for each of the conditions, in 1G and 2G we initially calculated the ratio between the recorded and linearly expected responses for individual neurons, and averaged the ration over all neurons. Following the reviewer’s comment we agree that this difference is confusing. In the revised manuscript we used a single unified method for calculating the averaged results. We preferred the more conventional method of averaging the response for each of the conditions (as used in 1F and 2F of the original manuscript). The new modified calculations are presented in Figure 2 and Figure 3 and in the subsection “Pairing optogenetic vM1 activation with passive whisker activation”.

*Is Figure 3 a single example neuron? If so, should a) make it clear that the response on the left and the response on the right are two individual neurons and b) the variability of the response across trials should be indicated.*

These are indeed responses from two different individual neurons, one responding to a ramp and hold passive whisker stimulation, and the other to artificial whisking against sandpaper. Following the reviewer’s comments we clarified this point in the revised manuscript (Figure legend 4). In addition, we added error bars to the averaged values.

*Figure 4 should also include measures of response variability. In Figure 5, the asterisk should be defined and exactly which groups were being compared should be stated clearly (e.g. in the Methods or in the legend or text). If the pooled controls are being compared to individual layers (i.e. the layers don't have their own separate control groups – which would not make sense but which is somewhat implied by the structure of the figure) a correction for multiple comparisons must be used.*

Following the reviewers’ comments we added error bars to the angular tuning experiments presented in Figure 5. As error bars are difficult to view in polar plots we present this data in a separate figure (Figure 5—figure supplement 1). Upon re-reading the manuscript, we agree that we were not clear about the statistical comparisons presented for the different cortical layers. For each layer we compared SI of the preferred angle with and without laser. This is now clarified in the figure legend of the revised manuscript (Figure legend 6).

*5) The authors should show some example histology that convinces the reader that the assignment to layers is accurate.*

We performed our experiments with a single shaft 16 contact silicone probe (NeuoNexus). The inter-contact distance in our electrodes was 50 microns. We categorized our recordings to putative layers based on the recording depth from pia (depth 0 was determined visually during electrode insertion, and the depth was calculated by the location of the electrode along the electrode shaft, assuming the silicone probe remained straight and unfolded). The recording location of each contact could not be confirmed histologically, and thus, we used the term putative cortical layer. Following the reviewer’s comment we further emphasized this points in the revised manuscript (in the subsection “Pairing optogenetic vM1 activation with passive whisker activation”).

*Reviewer #2:*

*The supralinear main effect observed in this paper is undeniably interesting.*

*Major:*

*I) Stimulus control is essential for this experiment:*

*1) Perpendicularly-arrayed bimorphs are notorious for ringing and distortion. I would like quantification of the amplitude and velocity of motion for all 8 directions for the piezo's used, and most importantly the ringing they demonstrated both for the first deflection and the other deflections in a train. Such quantification and description is essential.*

As pointed by the reviewer, distortions and ringing of the bimorphs during ramp and hold paradigm, especially in the case of two perpendicular bimorphs, are significant issues. Our lab has worked and published in the past on the issue (Garion et al., 2014 for ramp and hold stimuli and Lavzin et al., 2012 for angular tuning). In the previous version of the manuscript, we relayed on quoting these articles. However, following the reviewer's comment in the revised manuscript we detailed our procedure for eliminating ringing of the bimorphs (subsection “Whisker stimuli”). Specifically, passive ramp-and-hold stimulation the principle whisker was rapidly deflected (1300°/second) with a single ceramic piezoelectric bimorph for a period of 200 ms (Simons, 1983; Wilent and Contreras, 2004; Lavzin et al., 2012; Garion et al., 2014). To avoid ringing of the whisker during the rapid deflection phases we used a sigmoid function for the onset and offset piezo movements, resulting in an effective rise/fall duration of 20 ms. The deflections were controlled via a National Instruments board (PCI 6713), using custom routines written in MatLab. To monitor and calibrate whisker movements, and confirm lack of distortion or ringing of the stimulated whisker the deflection was monitored using a laser displacement sensor (LD1605-2; Micro-Epsilon OptoNCDT 1700) (Lotem and Azouz, 2009; Lavzin et al., 2012; Garion et al., 2014). Moreover, we confirmed our calibration, lack of distortions and lack of ringing by whisker tracking with a high-speed camera (1000 fps). For this procedure, whisking movement was photographed with a high-speed camera (Flare, 4M180MCL, 4 Megapixel, Dalsa Xcelera-x4-CL, IO industries at 1000 fps) and software (Streams 6; IO industries) with resolutions 600 × 350 pixels. Movement of full-length whisker was tracked semi-manually, and the angle and curvature of the whisker were calculated as described in Garion et al., 2014, using a homemade software written in MatLab (MathWorks, NA).

*2) I'm confused by the description of stimulation in the Results (200 ms ramp and hold stimuli) and in the Methods (100 ms).*

In all our experiments we used a 200 ms ramp and hold stimulation. The "on response" was quantified by recording the spike count during the initial 100 ms of the ramp and hold stimulation, and the "off response" was quantified by the spike count during the initial 100 ms after termination of the ramp and hold stimulus. In one instance in the Methods section, we mistakenly wrote the ramp and hold stimulus lasted only 100 ms (angular tuning). We corrected this point in the revised manuscript. The stimulation parameters are described in the subsection “Whisker stimuli”.

*II) I am skeptical about the 50 micron MEAs from neuronexus finding any single units of demonstrable selectivity in layers II/III. I certainly am not impugning the author's basic credibility (I don't think they are lying!) but getting well-isolated units in those layers, or layer IV especially, with such probes is quite hard. Without the virtue of a tetrode configuration or the use of smaller diameter contacts the claim of single unit identification is particularly suspect.*

No doubt, technical issues are critical for single unit analysis. I would like to address these concerns along two parallel routes:

1. In our recordings we used NeuroNexus 177 µ^2^ electrodes, which are the smallest contact size electrodes by NeuroNexus. We previously used 2-5 mega Ohm commercial tungsten electrodes, and our experience is that the NeuroNexus electrodes are superior in quality. For our spike sorting we used stringent spike sorting criteria for spike sorting based on the commercial spike sorting by the OFS offline spike sorting software from Plexon. Specifically, after thresholding of the raw filtered data (>3.5 SD of the baseline) sorting was performed by semi-automatic algorithms later verified and corrected manually, if necessary. Clusters were accepted as single unit if all the following criteria were met: 1) The waveform shape remained consistent and stable throughout recording (verified by the "Sort- Quality Vs. Time" analysis in the OFS software). Units were also excluded in case the average amplitude or half width of unit changed significantly (ANOVA test) between the first and last 20% of recorded spikes. 2) The firing rate was >0.5 Hz to allow for adequate sampling. 3) The inter-spike interval (ISI) was >2 ms to reflect the absolute refractory period of neurons.

4) The ISI distribution showed a smooth exponential-like curve. 5) Finally and most importantly statistical criterion of p<0.05 (multivariate ANOVA) of cluster separation. These points were added to the revised manuscript (subsection “Electrophysiological unit recordings”).

2. To further tackle the technical challenges associated with single unit sorting we repeated all our analysis for multi-unit data (threshold >3.5 SD above baseline). We found the similar to single unit analysis, multi-unit analysis also showed that vM1 optogenetic activation both supra-linearly amplified the responses of whisker stimulation, as well sharpened angular tuning. Thus, our multi-unit data verified the single unit results. The multi-unit data was added to the revised manuscript (subsections “Electrophysiological unit recordings”, third paragraph, “Pairing optogenetic vM1 activation with passive whisker activation”, fifth paragraph and “The effect of vM1 activation on angular tuning of neurons in the vS1 barrel cortex”, third paragraph, Figure 2 and Figure 6). Note that in the case of angular tuning we also added date from 6 additional 90-100 day old mice.

In addition, following the reviewer’s comment (as well as other reviewers) we added examples raw data traces (Figure 1—figure supplement 1), as well as raw clustering data (Figure 1—figure supplement 2).

*I also found the following criteria to feel a bit arbitrary and wasn't sure what it meant:*

*"For our analysis, we excluded all unstable neurons, with stability determined using two parameters. First, the baseline pre-stimulus frequency. Second, for every experimental paradigm at least one of the stimulation conditions was repeated twice during the experiment, and neurons in which the responded differentially to the two stimulation blocks were excluded."*

*First, what does the phrase "the baseline pre-stimulus frequency" have to do with stability? Second, multi-unit recordings can easily show run-by-run stability and, in fact, are potentially more stable than units for such recordings.*

In hindsight, we agree with the reviewer that we did not sufficiently define our inclusion and exclusion criteria. For the revised manuscript, we better defined criteria for including units in our analysis, and re-analyzed all our data according to these new and better-defined criteria. The criteria for spike sorting and stability of units is fully described in our response to a previous comment by the reviewer.

*So, I'd like to see the following to give me confidence:*

1) Application of autocorrelogram exclusion criteria.

We performed an auto-correlogram for each unit and excluded all units, which showed a response at ± 2 ms.

*2) Quantification of spike feature stability across the entire run (whether or not the response properties changed), and a discussion of which spike features were used.*

We monitored the amplitude and shape of the waveform of each sorted unit using the "Sort-Quality Vs. Time" analysis in the OFS software. In addition, units were excluded in case the average amplitude or half width of unit changed significantly (ANOVA test) between the first and last 20% of recorded spikes.

*3) Demonstration of the cluster quality for some number of units – I don't want to make this infinite, but as I said I'm skeptical, so at least a fair number of examples of stability in layers II/III units would be appreciated (say, 10 such examples from 10 experiments or so).*

Following the reviewer’s comment we added 10 individual examples (10 electrodes with 26 units), in which both cluster quality is demonstrated. This data is presented in Figure 1—figure supplement 2 of the revised manuscript. In addition, we present raw traces of our recordings (Figure 1—figure supplement 1 of the revised manuscript).

*Moderate:*

*I) In the 56 neurons recorded in vMI, what was the latency to firing following optogenetic stimulation, and was this a bimodal distribution (implying directly opto activated and synaptically opto activated populations).*

Following the reviewer’s comment we measured the latency between laser onset (10 ms pulse) and a significant (P<0.05) increase in firing. On average, we found the latency between laser onset and increased firing in M1 to be 4.81 ± 0.43 ms (mean ± SEM, 109 neurons from 5 rats, note we performed an additional experiment and increased the number of recorded neurons). Of the recorded neurons, 41% showed a mono-modal response, 43% showed a bi or multi-modal response, and 16% did not respond to the optogenetic stimulus. This data was added to the second paragraph of the subsection “Pairing optogenetic vM1 activation with passive whisker activation”.

*II) The authors should quantify the relative effect of stimulation on the optimal direction angle versus the worst direction angle, whatever these might have been. Obviously, there must be enhancement on average across randomly associated directions or else the main effect of Figure 1 couldn't stand: The sharpening obviously suggests, though, a significant different between best and worst, etc.*

As suggested by the reviewer we compared the effect of vM1 activation on the selectivity to the best and worst directions. To do so we calculated the selectivity index for the preferred direction and the worst three directions. vM1 activation increased the SI to the preferred direction (as also shown in the original manuscript). In contrast, optogenetic activation of vM1 decreased the SI to the worst directions (average of the worst 3 directions). The reduction in the SI occurred despite an absolute increase in the response amplitude to both the preferred and worst angular directions. This data is presented in the second paragraph of the subsection “The effect of vM1 activation on angular tuning of neurons in the vS1 barrel cortex” and Figure 6—figure supplement 1 of the revised manuscript.

*Reviewer #3:*

*[…] The manuscript would benefit, however, from a more detailed and extensive presentation of (raw) data and a more in-depth treatment of the crucial issue of relative input timing. Find my specific comments below.*

*1) Introduction, last paragraph: Why only 'partially disconnected'? Isn't the motor arm entirely uncoupled? Was this actually verified by confirming the absence of whisker movements following vM1 activation after the nerve cut?*

We indeed confirmed the location of vM1 by whisker movements induced by vM1 optogenetic stimulation in rats with intact buccolabial nerve. We further confirmed lack of whisker movements by vM1 optogenetic stimulation after cutting the buccolabial nerve. We added this information to the revised manuscript (subsection “Viral vector injection and optogenetic stimulation”, last paragraph).

In our original manuscript we used the wording "To partially dissociate the vibrissae sensory-motor loop" as there are connections between the sensory and motor limbs at the cortical, subcortical and brainstem levels, while we only severed the peripheral motor nerve. Yet following the reviewer's comment, we see that these wording may be confusing, and we omitted the word "partially" from the revised manuscript.

*2) Subsection “Electrophysiological unit recordings” and figures. The authors should provide some raw electrophysiology data, so that one can better judge the quality of data. What were the noise levels? How well could spike sorting be performed and on how many channels? How well could spike waveforms be separated and were there any putative fast-spiking units present? Do the reported spike rates represent changes in spike rate? It would also be interesting to see the LFP responses for the paired stimulation paradigms.*

No doubt, these points are very important, and I would like to address each point separately:

1) Following the reviewer’s comments we added filtered (1-5 KHz) raw traces (Figure 1—figure supplement 1), as well as examples of sorting in 10 individual electrodes with 26 individual units (Figure 1—figure supplement 2).

2) Based on the SNR definition of Rousche and diamond, 1999, typically, our unit's SNR were in the range 7-15, and the average was 11.2 ± 0.4 (mean

± SEM). This data was added to the revised manuscript (subsection “Electrophysiological unit recordings”, third paragraph).

3) Our recording electrode contained 16 channels of which we typically analyzed 13-15 channels.

4) We greatly elaborated on our spike sorting procedures, and clarified our inclusion and exclusion criteria for units (subsection “Electrophysiological unit recordings”, for details see response to the next comment by the reviewer).

5) In our recordings high resolution sorting can be performed, and the spike waveforms could be very well separated (for example see Figure 1—figure supplement 1 and Figure 1—figure supplement 2, and response to the next comment).

6) We did not analyze data for fast spiking neurons. We believe that with the introduction of optogenetics, solely classifying neurons according to their waveform and firing characteristics is not sufficiently accurate.

7) To tackle with the inherent technical issues associated with single unit spike sorting we repeated our analysis on multi-unit activity (events with an amplitude >3.5 SD of the baseline). The multi-unit analysis confirmed the results with single unit activity, and also showed that vM1 activation amplified the response to ramp and hold whisker activation (Figure 2 and subsection “Pairing optogenetic vM1 activation with passive whisker activation”, fifth paragraph), and sharpened angular tuning (Figure 6 and subsection “The effect of vM1 activation on angular tuning of neurons in the vS1 barrel cortex”, third paragraph).

8) Our spike count are an increase in the spikes above baseline (Spike count response-Spike count baseline). This is further explained in the Methods section of the revised manuscript.

*3) Furthermore: what were the respective baseline spike frequencies and why (and how) were these used to exclude data from the analysis (subsection “Electrophysiological unit recordings”, third paragraph). I also do not understand the motivation for excluding units that did not show a consistent response pattern upon repeating the stimulation paradigm. What type of responses did these cells actually show, in how far did they deviate from a nonlinear summation results? Why should one set such bias?*

Upon re-reading the manuscript, we also saw that our inclusion and exclusion criteria of units were not sufficiently clear. For the revised manuscript, we better defined criteria for including units in our analysis, and re-analyzed all our data according to these new and better-defined criteria.

Specifically, we initially recorded the multi-unit activity by recording events with an amplitude >3.5 SD above baseline from the filtered raw data (1-5 KHz). From this data we next sorted single unit activity using the OFS offline spike sorting software from Plexon. Sorting was initially performed by semi- automatic algorithms, and later verified and corrected manually. We accepted clusters as single unit if all the following criteria were met: 1) The waveform shape remained consistent and stable throughout recording. This was verified by the "Sort-Quality Vs. Time" analysis in the OFS software. Moreover, we excluded unstable units in case the average amplitude or half width of unit changed significantly (ANOVA test) between the first and last 20% of recorded spikes. 2) The firing rate was >0.5 Hz to allow for adequate sampling. 3) The inter-spike interval (ISI) was >2 ms to reflect the absolute refractory period of neurons. 4) The ISI distribution showed a smooth exponential-like curve. 5) Finally and most importantly statistical criterion of p<0.05 (multivariate ANOVA) of cluster separation.

These points were added to the revised manuscript (subsection “Electrophysiological unit recordings”). In addition to the revised manuscript, we added examples raw data traces (Figure 1—figure supplement 1) as well as raw clustering data (Figure 1—figure supplement 2). Following the reviewer comments (as well as of the other reviewers) in the revised manuscript, we no longer used the pre-stimulus firing as a criteria for including units in our analysis, but rather included all units with stable sorting and amplitude parameters.

Finally to further tackle the technical challenges associated with single unit sorting we repeated all our analysis for multi-unit data (threshold >3.5 SD above baseline). Our multi-unit data verified the single unit results. The multi-unit data was added to the revised manuscript (subsection “Pairing optogenetic vM1 activation with passive whisker activation”, fifth paragraph and subsection “The effect of vM1 activation on angular tuning of neurons in the vS1 barrel cortex”, third paragraph, Figure 2 and Figure 6).

*4) Artificial whisking paradigm (subsection “Whisker stimuli”, Figure 2, Figure 3). How exactly was the optogenetic stimulation paired with motor activation pulses? Was it a 20-ms light pulse at the beginning of the 10 pulses to the nerve?*

First, we want to point out that the artificial whisking experiments served mainly as a control for the ramp and hold stimulation. As a result, we wanted to keep stimulation conditions of vM1 as similar as possible to the ramp and hold protocol.

With respect to the timing and duration of vM1 activation, for the experiment described in Figure 2 of the original manuscript (Figure 3 of the revised manuscript) we used 20 ms optogenetic pulses, which were applied at the beginning of artificial whisking (10 pulses at 100 Hz for the protraction phase). It is true that the optogenetic pulse only covered 20% of the protraction phase. However, we found that even these conditions were sufficient to supra- linearly amplify the response in the barrel cortex. In Figure 3 of the revised manuscript we clearly marked the timing of vM1 whisker activation, buccolabial nerve stimulation pulse and protraction and retraction phase (see below).

In the experiments described in Figure 4 of the revised manuscript (Figure 3 of the original manuscript) we also applied 20 ms optogenetic pulses, but at different time intervals from whisker stimulation. 0 ms time interval meant that whisker stimulation and the optogenetic stimulation were initiated simultaneously, -20 ms meant that the optogenetic stimulation was initiated 20 ms before whiskers were stimulated, and so on.

*How large were the whisker protractions induced by this protocols?*

Our artificial whisking typically resulted in a 45-60° protraction. We have described our protocol, including a movie showing whisker movement in Garion et al., 2014.

*The spiking response appears only 20-ms long, how come? What happened in the remaining 80 ms of protraction/retraction cycle? In the eighth paragraph of the subsection “Pairing optogenetic vM1 activation with passive whisker activation” it is stated that spikes were evoked during protraction and retraction phase, but I can't see that. More details are needed here.*

In the original manuscript, we only showed a fraction of the protraction phase. In Figure 3 of the revised manuscript, we show a full protraction and retraction cycle, and the timing of buccolabial nerve stimulation and vM1 optogenetic stimulation. Typically, the protraction phase shows a short response; possibly due to the fact whiskers reach their final position before the end of the 100 ms protraction stimulation (for details see Garion et al. 2014).

*5) Subsection “Viral vector injection and optogenetic stimulation”, first paragraph. How precisely was the vM1 hit with the injections and the light stimulation? Was this verified by measuring whisker movements (or other movements?) by optogenetic vM1 stimulation before the facial nerve was cut?*

We localized vM1 by published coordinates, and confirmed the location of vM1 by monitoring whisking movements during optogenetic stimulation. We found the optogenetic stimulation of vM1 evoked whisking movements in rats with intact uncut buccolabial nerves. This data was added to the last paragraph of the subsection “Viral vector injection and optogenetic stimulation”.

*6) I find the timing experiments very interesting but I am not sure how to interpret the peak revealed at -20 ms. This is a time scale where the conduction delays etc. play an important role. So how was the timing exactly defined (onset of piezo drive and onset of LED illumination, I presume)?*

In these experiments vM1 was stimulated with 20 ms laser pulses. We chose the 20 ms pulse after examining the effect to 5-20 ms laser pulse duration (Figure 1). The relative timing between vM1 stimulation and whisker stimulation was defined as T=onset time of laser pulse-onset time of whisker stimulation (see response to previous comment).

*Was there any dead time for mechanical stimulation of the whisker considered? In particular, how was the timing defined for the artificial whisking stimulus? Were the whisker movements (presumably stick-slip events) monitored and their timing analyzed? Were any axonal conduction times and synaptic delays taken into account? Obviously, these questions are important to understand what the real timing difference at the integrating neurons in barrel cortex might be. Supplementary whole-cell recordings could be very helpful here, to quantify when exactly inputs from both pathways actually arrive with these stimulation paradigms.*

The points raised by the reviewer are of course very valid and important. In our experiments we only controlled for the timing stimulation was triggered, but not for synaptic or transmission delays, nor for the timing of stick and slip events during artificial whisking. For that reason we used relatively long optogenetic stimulation (20 ms), and concentrated on the on response and initial protraction phase (90-100 ms). We further stressed this point in the last paragraph of the subsection “Temporal roles governing the interactions between vM1 and whisker sensory inputs”.

*7) Angular tuning in barrel cortex apparently is a complicated matter and among other things seems to especially depend on age (Kremer study). In addition, the housing conditions (use of whiskers) may affect the outcome. The age of rats used here is just in between the ages when no angular tuning was observed and when it later was established. I find the examples in Figure 4 not convincing, as the pure whisker-evoked responses appear relatively untuned and the responses enhanced by optogenetic vM1 activation mostly display multiple peaks (often in orthogonal directions). Cells from what layer are actually shown in Figure 4? It might be helpful to show distributions of the absolute SI values for the different stimulation protocols.*

Following the reviewer’s comments we:

1) Performed additional experiments in which we repeated our angular tuning experiments in 90-100 day old rats (n=6). These experiments again showed sharpening of angular tuning by vM1 optogenetic activation. Interestingly we saw no significant difference in the control angular tuning between 50-60 day old and 90-100 day old rats. This new data is presented in Figure 5, Figure 5—figure supplement 1, Figure 6, Figure 6—figure supplement 1 and in the second paragraph of the subsection “The effect of vM1 activation on angular tuning of neurons in the vS1 barrel cortex”.

2) In Figure 4 of the revised manuscript, we now show four examples from the original set of experiments (50-60 day old) and two additional examples from the new experiments (90-100 day old). In addition, we added the putative layers of the recorded neurons.

3) We believe some of our examples reveal angular tuning, which are consistent with previously published data on the phenomenon, including the existence of more than one peak in the angular tuning curve (See Bruno et al., 2003; Andermann and Moore, 2006). We also show an example of a unit that showed minimal angular tuning prior to opto-stimulation, and transformed into an angular tuned neuron during opto-stimulation.

4) As suggested by the reviewer we added a histogram of the SI values with and without laser activation (Figure 6 of the revised manuscript), as well as the effect of vM1 activation on the SI of both preferred direction and worst three directions (Figure 6—figure supplement 1).

[Editors' note: the author responses to the re-review follow.]

*Essential revisions:*

*1) One reviewer states: "I appreciate the addition of details on how the stimulator was calibrated-these are all the right approaches and tools. I can guarantee that using these parameters, that unless they use a specific compensatory algorithm, they almost certainly cannot get ringing under 5% of stroke magnitude, and I doubt it can be under 10%. I want the following included in the final manuscript → An actual analysis of the mean amplitude of the ringing. Saying you did not see it does not reflect quantification of the effect. Say how large in angle of vibrissal base motion and at what frequency the ring is on average across trials and across the different stimulators used." The reviewer notes that this is likely to have an effect on the direction selective responses.*

Ringing of piezo bimorphs is indeed significant problem. Regarding this technical issue, we wish to note that we have already published two papers with the same stimulation system in two very distinguished journals, *eLife* and Nature. Thus, while writing the previous version of the manuscript that our description of our stimulation protocol combined with quotations of our prior papers would be sufficient.

Following the reviewer's comments and requests we have further elaborated on the technical details of our ramp and hold stimulation protocol, and added experimental data regarding the movement and ringing of the piezo bimorphs during ramp and hold stimulation, as monitored with a high-speed camera (1000 fps). In the revised manuscript we (1) present the formula by which we smoothened the onset and offset of the ramp and hold stimulation pulse. (2) We show single traces of the piezo bimorph movements during ramp and hold pulses smoothened to different degrees. (3) Finally, we present the averaged ringing amplitude of the piezo bimorphs at different degrees of smoothening of the ramp and hold pulses (different smoothening factors).

This data is presented in Figure 1—figure supplement 3 and in the subsection “Vibrissa stimuli”. As shown by the data we succeeded in minimizing ringing in our experimental conditions.

*2) A second reviewer felt that the documentation of the degree of direction selectivity was inadequate. They did not feel the statement "Consistent with previous results we found angular tuning in neurons.…" (subsection “The effect of vM1 activation on angular tuning of neurons in the vS1 barrel cortex”, first paragraph) was well supported. They felt that the criteria for when a cell's response is considered well tuned were not clear and that the SI as calculated made comparisons across neurons with very different response amplitudes difficult. It is suggested that the authors "confirm their interpretation with an alternative, more robust analysis method of direction-tuning, based on the mean response vector not the Rmax (Kremer et al. 2011; Mazurek et al., Front Neural Circ 2014). And provide a statistical argument for their statement 'we found angular tuning'."*

Following the reviewer's comments, we performed further analysis and quantification of angular tuning. As eloquently presented by Mazurek et al., 2014, quantification of selectivity is a complex issue. In the original manuscript we chose the SI as our main quantification parameter. We introduced two additional parameters:

1) Angular tuned neurons, as defined by a significant difference at the 0.05 level for comparison of the responses to the preferred angle and to the three least preferred angles. We found that about 60% of S1 neurons showed angular tuning, with no significant differences between putative layers 2-5. This data was added to the subsection “The effect of vM1 activation on angular tuning of neurons in the vS1 barrel cortex”.

2) Vector sum analysis for angular tuning. We used a second parameter for quantifying angular tuning, the vector sum. In turn, we used the vector sum to examine the effect of vM1 activation on angular tuning. We found that similar to SI analysis optogenetic activation vM1 significantly increased the amplitude of the summed vector. This data was added to Figure 6 (panel 6C) and at the end of the subsection “Vibrissa stimuli” and “The effect of vM1 activation on angular tuning of neurons in the vS1 barrel cortex”.

*3) One of the reviewers also notes that "laser stimulation of vM1 alone in essentially all cases did not evoke spiking activity at all (except perhaps for Figure 1?). Thus, the 'supralinear' effect essentially consists in an upregulation or facilitation of the sensory-evoked response in vS1, which could be simply explained by additional pre-depolarization mediated by the M1-to-S1 projections, given vM1 is stimulated at the right time briefly before the sensory stimulus. While the authors mention this simple explanation (vM1 projection fibers helping vS1 neurons to reach the – nonlinear – spike threshold), they only refer to it as 'additional cellular mechanism' in the third paragraph of the subsection “Possible mechanisms underlying vM1 mediated response amplification and sharpening of angular tuning in vS1 barrel neurons”. Their primary 'attractive potential mechanism' of dendritic amplification (in the first paragraph of the aforementioned subsection) in my view is, however, largely speculative." The authors should consider toning down their use of 'supralinear responses' and may consider using terms like 'facilitation' or 'modulation.' At the very least they should give more equal weight to the simplest interpretation of these effects.*

We agree with the reviewer that "simple" effects on the axonal initiation zone can explain our findings, and have discussed this possibility in the previous version manuscript. Yet we feel that "simple effect" of vM1- mediated EPSPs on the axonal non-linearity of vS1 neurons is less likely to explain the large effect we found. However, as we did not examine this issue experimentally, and following the reviewer's comments, we down toned the dendritic amplification scenario in the revised manuscript.